# Wnt1 silences chemokine genes in dendritic cells and induces adaptive immune resistance in lung adenocarcinoma

Dimitra Kerdidani [1,2], Panagiotis Chouvardas [1,3,4], Ares Rocanin Arjo[5], Ioanna Giopanou [6], Giannoula Ntaliarda[6], Yu Amanda Guo[7], Mary Tsikitis [8], Georgios Kazamias [9], Konstantinos Potaris[10], Georgios T. Stathopoulos [6,11], Spyros Zakynthinos[2], Ioannis Kalomenidis [2], Vassili Soumelis [5], George Kollias [1,12] & Maria Tsoumakidou [1]

Lung adenocarcinoma (LUAD)-derived Wnts increase cancer cell proliferative/stemness potential, but whether they impact the immune microenvironment is unknown. Here we show that LUAD cells use paracrine Wnt1 signaling to induce immune resistance. In TCGA, Wnt1 correlates strongly with tolerogenic genes. In another LUAD cohort, Wnt1 inversely associates with T cell abundance. Altering Wnt1 expression profoundly affects growth of murine lung adenocarcinomas and this is dependent on conventional dendritic cells (cDCs) and T cells. Mechanistically, Wnt1 leads to transcriptional silencing of CC/CXC chemokines in cDCs, T cell exclusion and cross-tolerance. Wnt-target genes are up-regulated in human intratumoral cDCs and decrease upon silencing Wnt1, accompanied by enhanced T cell cytotoxicity. siWnt1-nanoparticles given as single therapy or part of combinatorial immunotherapies act at both arms of the cancer-immune ecosystem to halt tumor growth. Collectively, our studies show that Wnt1 induces immunologically cold tumors through cDCs and highlight its immunotherapeutic targeting.

[1] Division of Immunology, Biomedical Sciences Research Center Alexander Fleming, Vari-Athens 16672, Greece. [2] 1st Department of Critical Care and Pulmonary Medicine, Medical School, National and Kapodistrian University of Athens, Athens 10676, Greece. [3] Department of Medical Oncology, Inselspital, Bern University Hospital, University of Bern, Bern 3012, Switzerland. [4] Department for BioMedical Research, University of Bern, Bern 3012, Switzerland. [5] Integrative Biology of Human Dendritic Cells and T Cells, Institute Curie, Paris 75005, France. [6] Laboratory for Molecular Respiratory Carcinogenesis, Department of Physiology, Faculty of Medicine, University of Patras, Rio, Achaia 26504, Greece. [7] Computational and Systems Biology, Genome Institute of Singapore, Agency for Science Technology and Research, Singapore 138672, Singapore. [8] Center of Basic Research, Biomedical Research Foundation of the Academy of Athens, Athens 11527, Greece. [9] Department of Histopathology, Evangelismos General Hospital, Athens 10676, Greece. [10] Department of Thoracic Surgery, Sotiria General Hospital, Athens 11527, Greece. [11] Comprehensive Pneumology Center (CPC) and Institute for Lung Biology and Disease (iLBD), Ludwig-Maximilians University and Helmholtz Center Munich, Member of the German Center for Lung Research (DZL), Munich, Bavaria 81377, Germany. [12] Department of Physiology, Medical School, National and Kapodistrian University of Athens, Athens 11527, Greece. Correspondence and requests for materials should be addressed to M.T. (email: tsoumakidou@fleming.gr)

The canonical (b-catenin-dependent) Wnt pathway is key to healthy tissue homeostasis and to the increased cancer cell proliferative, metastatic and stemness potential[1]. Although activating mutations in intracellular components of the pathway induce Wnt ligand-independent signaling in cancer cells, the importance of ligand-dependent signaling is increasingly appreciated[2]. Targeted therapies against Wnt ligands show good preclinical responses and are tested in human trials[3]. A major drawback of the available treatments is that they non-specifically target groups of ligands and receptors and are associated with a high frequency of adverse events[3]. Blocking cancer-specific single Wnts should be a safer and more efficient approach. Unfortunately, there are 19 human Wnts, multiple points of intersection and crosstalk connecting the various Wnt signaling cascades and little evidence for the existence of specific Wnt ligands with non-redundant roles in cancer[1].

Adding another level of complexity to the Wnt/b-catenin pathway is that it is among few oncogenic pathways found to impact adaptive immunity, as shown in melanoma[4–8]. B-catenin activation in melanoma cells impedes CCL4 production via ATF3 upregulation, preventing intratumoral migration of CD103[+] conventional dendritic cells (cDCs)[4]. CD103[+] cDCs are pivotal for tumor immunosurveillance: (i) they transport tumor antigens to regional lymph nodes, where they cross-prime T cells[9] and (ii) they are key cellular sources of the T cell-attracting chemokines at tumors[10]. In addition to the cDC-exclusion effect of melanoma cell-intrinsic b-catenin activation, paracrine Wnt5a signaling from melanoma cells to DCs leads to b-catenin activation, tolerogenic gene transcription, as well as fatty acid oxidation and post-translational activation of the immunosuppressive enzyme indoleamine (IDO)[11–14]. Recent data point to a more universal link between Wnt/b-catenin activation and T cell exclusion across most major human cancers[15]. T cytotoxic cell abundance is an important prognostic cancer biomarker, highlighting the translational value of these findings[16]. Considering that Wnt5a mainly works through b-catenin independent pathways and also exhibits tumor-suppressive functions in certain cancers[17], other Wnts that have yet to be discovered besides Wnt5a may drive T cell exclusion through different mechanisms outside melanoma.

Lung cancer is the world's leading cause of cancer death (Available from: http://www.who.int/mediacentre/factsheets/fs297/en/). Approximately 40% of all the diagnosed cases are lung adenocarcinomas (LUADs). Canonical Wnt ligand-producing niches drive a stem-like phenotype in LUAD and genetic perturbation of Wnt production or signaling suppress tumor progression[2]. Whether there are any immunosuppressive functions of LUAD-secreted Wnts is unknown. This is of paramount clinical importance because lung cancer cells express neoantigens that can trigger immunological responses, if unleashed from tumor-induced immunosuppression[18].

Unbiased analysis of the Cancer Genome Atlas (TCGA) transcriptomics database shows that amongst all human Wnts, Wnt1 correlates positively with the expression of tolerogenic genes across the vast majority of cancers, including LUAD. In a distinct cohort of human LUADs, Wnt1 inversely associates with T cell abundance. Ex vivo assays with primary human LUAD cells and models of lung adenocarcinoma show that Wnt1 impairs cross-priming of T cytotoxic cells and induces T cell exclusion from tumors via cDCs. Rather than impacting tumor cDC infiltration, Wnt1 acts paracrine on intratumoral cDCs to silence expression of chemokine genes. Wnt1 siRNA-loaded nanoparticles rescue intratumoral cDCs from b-catenin activation and act in synergy with DC-target therapies to halt LUAD growth.

## Results

**Upregulation of Wnt1 in immune resistant human LUADs**. To address which Wnt ligands can drive adaptive immune tolerance, we analyzed the gene expression directory of The Human Cancer Genome Atlas. An unbiased list of 23 well-established immuno-suppressive genes was created and their expression correlated to Wnt ligand expression across all cancers. All known Wnts were ranked according to their distributions of the ranks of mean correlations. Wnt1 showed the highest means of correlations as well as the lowest distribution of the means (Fig. 1a). Intriguingly, among all Wnts, Wnt1 is the strongest negative prognostic factor in LUAD[19–23]. So we proceeded to immunohistochemical analysis of an in-house biobank of primary human LUADs (Supplementary Table 1). Human LUAD Wnt1 mRNA levels (qPCR) correlated negatively to numbers of CD8[+] T cells and granzyme B[+] cytotoxic cells (Fig. 1b).

Having shown that Wnt1 is a marker of adaptive immune tolerance in human LUAD, we sought to investigate its relevance to other types of cancers. We focused on colon, breast, hepatocellular, gastric and renal clear cell carcinomas, due to their reported associations with Wnt1[24–28]. The abundance of CD8[+] T cells can be measured by expression of the signature genes CD8a and CD8b[29]. Therefore, we assessed the distribution of Wnt1-CD8a, Wnt1-CD8b correlations in the TCGA database, including paired tumor and tumor-free samples, by calculating the z-scores of the Spearman correlation values. We used LUAD as positive control. GTEx database was used in order to include an unrelated to TCGA control. LUAD was the only tumor type that the z-score was negative for both CD8a and CD8b (Supplementary Figure 1). Additionally, LUAD was the tumor type with the lowest Wnt1-CD8a correlation z-score. For colon adenocarcinoma, the lack of association between intratumoral T cytotoxic cells and Wnt1 gene expression seems to be confirmed by IHC-based CD8[+] T cell enumeration (Dr Tan Bee Huat Iain, personal communication). Albeit these preliminary results cannot safely rule out a negative correlation between Wnt1 and T cells in other tumors, they do suggest a more important role for Wnt1 in LUAD tolerance. This could be related to Wnt1 overexpression being particularly frequent in LUAD compared to other types of cancer.

**Wnt1 cross-tolerizes endogenous and transferred T cells**. In order to investigate the functional implications of the correlations noted above we undertook a series of experiments using the Lewis Lung (adeno)Carcinoma (LLC) cell line, which expresses several Wnt ligands, among which Wnt1 is moderately expressed (Supplementary Figures 2, 3). Human LUAD cells overexpress Wnt1[19,20,30], so we virally transduced LLC cells with a Wnt1-expressing vector. Wnt1-overexpressing LLC cells (Wnt1-LLC) showed only slightly increased levels of active b-catenin compared to control (Empty) LLC cells and no measurable cell-autonomous difference in their proliferation (Fig. 2a). After lung transplantation in syngeneic immunocompetent mice, Wnt1 protein levels (ELISA) were higher in tumors from Wnt1-LLC vs. control LLC (Supplementary Figure 4). Albeit Wnt1 over-expression did not confer a proliferation advantage in LLC cells in vitro, in vivo they grew faster to give larger tumors (Fig. 2b). They also tended to home and proliferate faster in the lungs upon intravenous administration (Fig. 2c). Qualitative and quantitative analysis of immune cell subsets showed a relative decrease in the adaptive immune compartment in Wnt1-overexpressing lung tumor, which affected predominantly the population of CD8[+] T cytotoxic cells (Fig. 2d, e, Supplementary Figure 5). Consistent with the importance of tumor-immune interactions in

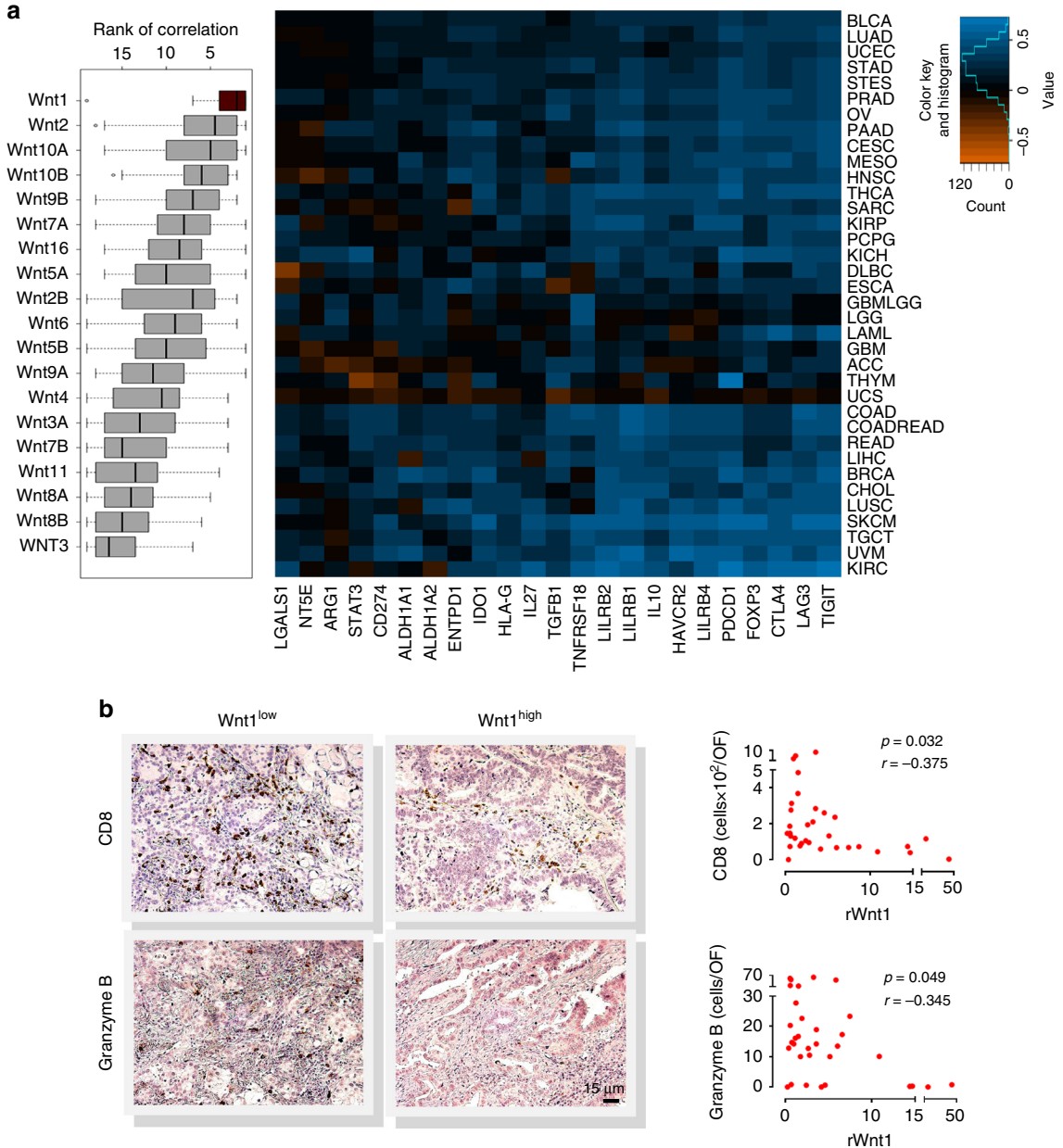

**Fig. 1** Upregulation of Wnt1 in adaptive immune resistant human tumors. **a** Distributions of the ranks of correlations between Wnt ligand gene expression and expression of 23 immunosuppressive genes across all cancer types of the TCGA dataset (>11,000 tumors) (Left) and heatmap representing color-coded Pearson correlations for Wnt1 (Right). Column side annotations are names of immunosuppressive genes and row side annotations are names of TCGA cancer types. **b** CD8[+] T cytotoxic cells and granzyme B[+] cells identified as brown stained cells by immunohistochemistry (IHC) in paraffin-embedded human tumor sections (Left). Scatterplots depicting the relationship between Wnt1 gene expression (RT-PCR) and numbers of CD8[+] and granzyme B[+] cells per tumor optical field (OF) (IHC). Pearson Correlation. (Right). Each point represents a single donor ($n = 33$). Source data are provided as a Source Data file

Wnt1-driven tumorigenesis, Wnt1-LLC cells grew faster in immunocompetent but not in immunodeficient RAG mice (Fig. 2f). Specific recognition of tumor antigens by CD8[+] cytotoxic T lymphocytes is one of the cardinal features of tumor immunosurveillance. Indeed, there was a striking decrease in numbers and activation status of intratumoral OVA-specific T cytotoxic cells upon inoculation of ovalbumin-expressing Wnt1-LLC cells, further suggesting that Wnt1 impacts T cell responses against cancer-specific antigens (Fig. 2g). To extrapolate our findings in another lung adenocarcinoma cell line, we transduced the Fula cells[31], which are derived from an autochthonous urethane-induced lung tumor, with Wnt1-expressing viral vector.

Implantation of Wnt1 overexpressing vs. control Fula cells in the lungs of syngeneic mice resulted in faster tumor growth and decreased CD8[+] T cell infiltration (Supplementary Figure 6).

To seek additional evidence that Wnt1 enhances LUAD tolerance, through stable genome engineering we generated LLC cells that expressed very low levels of Wnt1 (shWnt1 LLC cells) and compared them to control (Scramble) LLC cells. Wnt1 silencing slightly decreased b-catenin activation and impaired cancer cell proliferation (Fig. 2h), indicating that Wnt1 is required for cell-autonomous growth. However, shWnt1-OVA-LLC cells grew much slower in the lungs of immunocompetent mice vs. mice depleted of CD8[+] cells, while

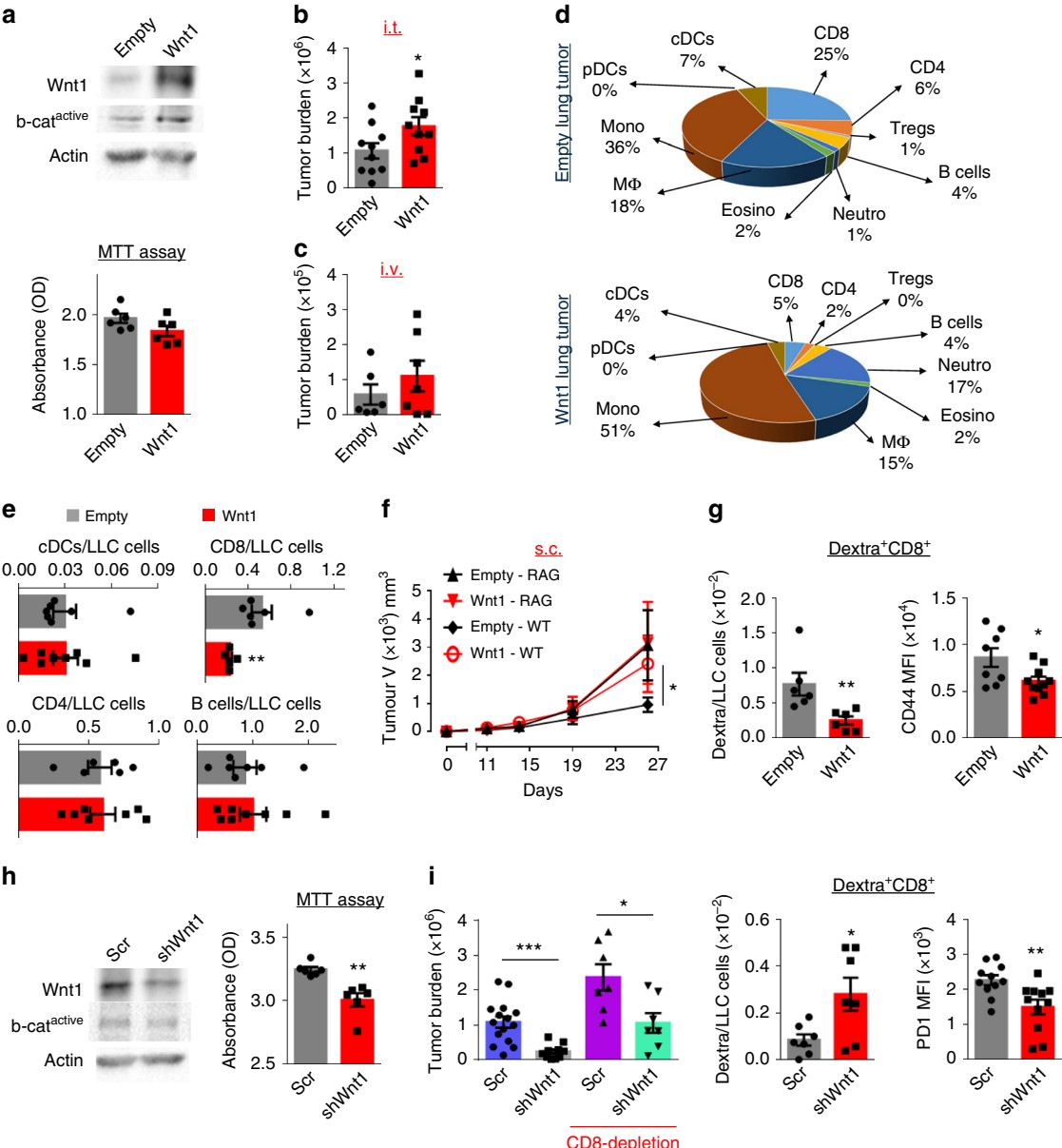

**Fig. 2** Wnt1 impairs adaptive immune surveillance in syngeneic models. **a** Expression of Wnt1 and active b-catenin (western blot) in LLC cells transduced with Wnt1 (Wnt1) or Empty (Empty) viral vectors (Up). In vitro proliferation (MTT assay) (Bottom). **b, c** Tumor burden (total absolute number of LLC cells) after intrathoracic (i.t.) implantation or intravenous (i.v.) administration. **d** Cellular profiles of lung tumors. Pies depict mean percentages among CD45+ cells. **e** Numbers of intratumoral cDCs, CD8 T, CD4 T, and B cells per cancer cell. **f** Tumor growth after subcutaneous implantation of ovalbumin (OVA)-LLC cells in immunocompetent vs. immunodeficient RAG mice. **g** Numbers of OVA-specific (dextramer+) T cytotoxic cells per cancer cell and T cell CD44 expression. **h** Expression of Wnt1 and active b-catenin (western blot) in LLC cells transduced with shWnt1 or Scramble viral vectors (left). In vitro proliferation (MTT assay) (right). **i** Tumor burden (total absolute number of OVA-LLC cells) after intrathoracic (i.t.) implantation in untreated vs. aCD8-treated mice (left). Numbers of dextramer+ T cytotoxic cells per cancer cell and T cell PD1 expression (Right). **a–i** Error bars represent mean with SEM. **b–g, i** Cell numbers and profiles were assessed by FACS. Data are representative or cumulative of at least two independent experiments with 5–9 mice per group. *$p < 0.05$; **$p < 0.01$, Mann–Whitney. Source data are provided as a Source Data file

lungs harboring shWnt1-OVA-LLC tumors were characterized by an increased number of OVA-specific T cytotoxic cells, with lower PD1 expression (Fig. 2i). These data collectively suggest that Wnt1 is a pro-tumorigenic signal in LUAD that acts prominently via adaptive immune-driven mechanisms.

To determine whether the Wnt1-driven adaptive immune resistance could be recapitulated in an adoptive T cell transfer model, we adoptively transferred OTI T cells to mice bearing already grown OVA-LLC lung tumors. Wnt1 overexpression rendered OVA-LLC tumors partially resistant to adoptive OTI

cell therapy (Fig. 3a). Comparative analysis of lung tumor-infiltrating OTI T cells on day 4 post adoptive transfer, showed decreased numbers but no significant differences in the expression of the effector cytokine IFN-γ (Fig. 3b). FACS staining showed that the vast majority of OTI T cells from Wnt1-LLC tumors expressed PD1 at much higher levels compared to those from control tumors (Fig. 3c). They also showed increased expression of other major inhibitory molecules, i.e., TIGIT and TIM-3 (Fig. 3d). Importantly, lung tumor-infiltrating OTI T cells of Wnt1 overexpressing tumors showed increased co-expression

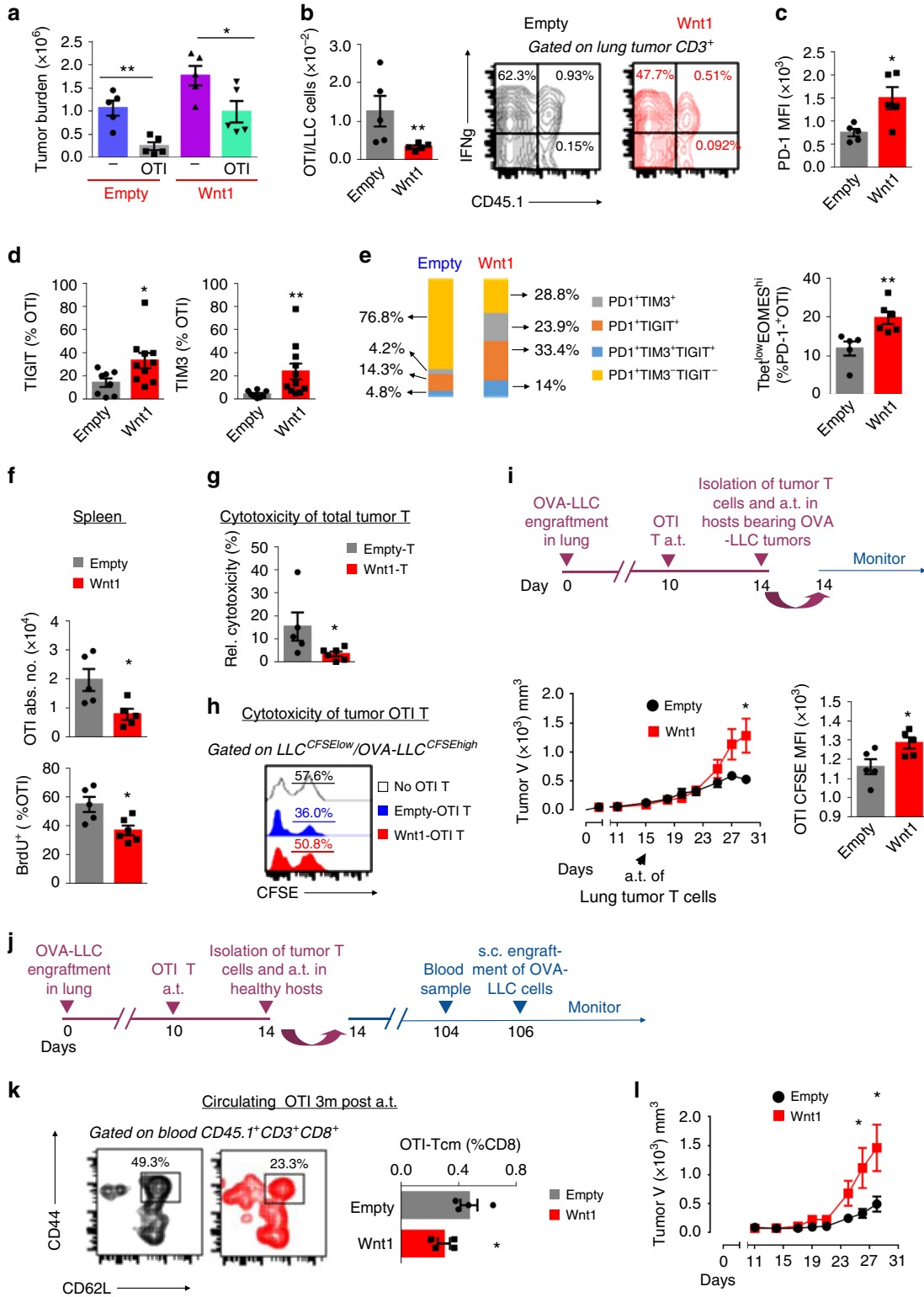

of PD1-TIGIT-TIM3 and a Tbet^neg EOMES^high profile, signs of severe exhaustion (Fig. 3e)[32]. To investigate whether Wnt1 afflicted proliferative responses of naïve OTI T cells we systemically administered BrdU to the mice on day 3 post adoptive transfer. On day 4 there were fewer total and BrdU positive splenic OTI T cells in Wnt1 overexpressing tumors (Fig. 3f).

To directly assess the anti-tumor cytotoxicity of intratumoral CD8+ T cells we isolated them 4 days post adoptive transfer. T cells from Wnt1 overexpressing lung tumors showed low cytotoxicity against OVA-LLC cells ex vivo (Fig. 3g). This was at least partly due to the defective function of OTI T cells, because purified intratumoral OTI T cells from Wnt1-OVA-LLC tumors showed low cytotoxicity against OVA-LLC cells (Fig. 3h). In

**Fig. 3** Wnt1 induces cross-tolerance of adoptively transferred transgenic T cells. **a** Tumor burden (total absolute numbers of Wnt1 overexpressing (Wnt1) vs. control (Empty) OVA-LLC cells) after intrathoracic (i.t.) implantation and adoptive transfer of OVA-specific OTI T cells. **b** Numbers of OTI T cells per cancer cell (left). IFN-γ expression among adoptively transferred (OTI) and endogenous CD8[+] T cells (Right). **c** PD1 expression by intratumoral OTI T cells. **d** Percentages of TIGIT and TIM-3 expressing intratumoral OTI T cells. **e** Mean PD1-TIGIT-TIM-3 co-expression depicted in bars (left) and percentages of severely exhausted Tbet[low]EOMES[high] cells among PD1[high] OTI T cells (Right). **f** Total absolute numbers of splenic OTI T cells (Up) and proliferation rates assessed by in vivo BrdU incorporation (Bottom). **g** OVA-LLC cells were co-cultured with intratumoral T cells sorted from Wnt1-overexpressing vs. control OVA-LLC tumors. Relative T cell cytotoxicity: cancer cell death in the presence relative to the absence of T cells. **h** Equal mixtures of CFSE-low LLC and CFSE-high OVA-LLC cells co-cultured with purified intratumoral OTI T cells from Wnt1-overexpressing vs. control OVA-LLC tumors. Histograms depict percentages of OVA-LLC cells among total LLC cells. **i** Mice bearing subcutaneous OVA-LLC tumors were adoptively transferred with purified CFSE-labeled OTI T cells from Wnt1-overexpressing vs. control OVA-LLC tumors (Up). Flank tumor growth (bottom left). OTI T cell proliferation measured by CFSE-dilution (Bottom right). **j** Healthy mice were adoptively transferred with purified OTI T cells from Wnt1-overexpressing vs. control OVA-LLC tumors. **k** FACS contour plots and graph depict peripheral blood OTI T memory cells 3 months post adoptive transfer. **l** Flank tumor growth after challenge of the adoptively transferred mice (as in **j**) with OVA-LLC cells. **a–l** Cell numbers and profiles were assessed by FACS. Error bars represent mean with SEM. Data are representative or cumulative of at least two independent experiments with 4–8 mice per group. *p < 0.05; **p < 0.01, Mann–Whitney or t-test (K). Source data are provided as a Source Data file

accordance to these in vitro findings, when we adoptively transferred intratumoral CD8[+] T cells from Wnt1-OVA-LLC tumors to recipient mice bearing established subcutaneous OVA-LLC tumors (secondary transfer), we observed low proliferation rates of intratumoral OTI T cells and enhanced tumor growth (Fig. 3i).

Various parameters influence the outcome of adoptive T cell therapy and T cell differentiation to memory cells is amongst the most crucial factors[33]. To investigate whether Wnt1 overexpression impairs the ability of the adoptively transferred OTI T cells to become memory cells we isolated intratumoral OTI cells 4 days post-primary adoptive transfer to lung tumor-bearing mice and transferred them to syngeneic healthy mice (secondary adoptive transfer) (Fig. 3j). Three months post the secondary transfer there were fewer CD44[+]CD62L[+] OTI T Central Memory (TCM) cells at the peripheral blood of mice that received cells from Wnt1-LLC tumors vs. control tumors (Fig. 3k). In accordance, when all mice were challenged with OVA-LLC cancer cells, tumors grew at much slower rates in the control mice (Fig. 3l), suggesting that impaired differentiation of OTI cells to memory cells is a Wnt1-driven mechanism of adoptive T cell therapy failure. Along with the fact that Wnt1 overexpression shifted the profile of adoptively transferred OTI T cells towards hypoproliferative, tolerogenic and exhausted and that we observed lower numbers of intratumoral OTI T cells and impaired cytotoxicity against OVA-LLC cells, these results strongly support the proposal that Wnt1 enables immunogenic tumors evade cross-priming of cancer antigen-specific CD8[+] T cells and turns tumors immunologically cold.

**Wnt1 overexpressing LUADs depend on cDCs to evade T cells**. Previous studies have shown that b-catenin activation renders cDCs tolerogenic[13, 34, 35]. We sought to characterize cDCs infiltrating Wnt1 overexpressing tumors. First, we quantified DC subsets, i.e. cDCs1, cDCs2, pDC, Langerhans cells, as well as monocyte-derived DCs (moDCs)[36]. Two independent experiments showed that Langerhans cells do not infiltrate orthotopic LLC tumors. The other 4 subsets, i.e. cDCs1, cDCs2, pDCs and moDCs, were consistently identified, with cDCs1 and pDCs being particularly scarce (Supplementary Figure 7). No differences were detected between Wnt1 overexpressing and control tumors in either experiment (Supplementary Figure 7). Gene expression analysis of intratumoral cDCs revealed that they expressed several members of the Frizzled family of receptors and among those expressed Frizzled 1 was reported to ligate to Wnt1 in the String database (Supplementary Figure 8). There are also many pleiotropic functions of the Wnt/b-catenin pathway reported in T cells, which are known to express several Frizzled receptors[37, 38].

We addressed whether our findings were mediated via Wnt1 acting directly on T cells or indirectly through DCs. Active b-catenin was increased in intratumoral cDCs, but not in CD8[+] T cells of Wnt1-LLC lung tumors, arguing in favor of an indirect effect of Wnt1 on T cells via cDCs (Fig. 4a). Active b-catenin did not differ between intratumoral cDCs1 and cDCs2 of Wnt1-LLC tumors (data not shown), suggesting similar levels of Wnt pathway activation on both subsets. Axin2 is a principal b-catenin target gene. In Axin2 reporter mice with LLC lung tumors of variable Wnt1 expression, Axin2 expression was lowest in cDCs of Wnt1-silenced tumors and highest in cDCs of Wnt1-overexpressing tumors (Fig. 4b).

Wnt proteins may signal distantly upon release by solubilization[39], formation of exosomes[40] or loading on lipid-protein particles[41]. We investigated whether Wnt1-induced signaling might occur in regional lymph nodes. We assessed Wnt pathway activation in mesothoracic lymph node cDCs of Axin2 reporter mice. We detected no differences between mice bearing Wnt1 overexpressing or control lung tumors (Supplementary Figure 9). We conclude that Wnt1-induced signaling rather occurs in the tumor microenvironment than in lymph nodes.

To strengthen our data we explored whether ex vivo exposure of cDCs to rWnt1 can induce signaling on cDCs. We acquired a commercially available rWnt1 protein and recapitulated experimental conditions under which rWnt3a triggers signaling[42]. rWnt1 succeeded to activate b-catenin in purified splenic cDCs (Supplementary Figure 10). To substantiate further Wnt1 paracrine signaling in cDCs, we exposed splenic cDCs in culture supernatants of several Wnt1 overexpressing cancer cells vs. those of control cells. We consistently observed b-catenin activation upon exposure to Wnt1 cell-derived vs. control supernatants (Supplementary Figure 10). Taken together these data, they strongly support the link between paracrine Wnt1-signaling and activation of b-catenin in DCs.

To investigate whether the Wnt1-driven tumor-promoting properties actively depended on cDCs, we used the zDC-DTR mice, which constitutively express the diphtheria toxin receptor in DCs of the myeloid lineage[43]. Wnt1-OVA-LLC did not grow larger in mice depleted of cDCs (Fig. 4c). These data collectively identify cDCs as the primary mediators of repressive Wnt1 signaling.

To further substantiate the role of cDCs we proceeded to intratumoral cDC purification and analyses, using RAG mice as hosts, as they had shown minor differences in tumor growth upon inoculation of Wnt1-LLC or Empty-LLC cells (Fig. 2f). cDCs purified from Wnt1-OVA-LLC tumors (Wnt1-cDCs) did not stimulate efficiently the TCRs of naive OTI T cells and induced low granzyme B production (Fig. 4d). Accordingly,

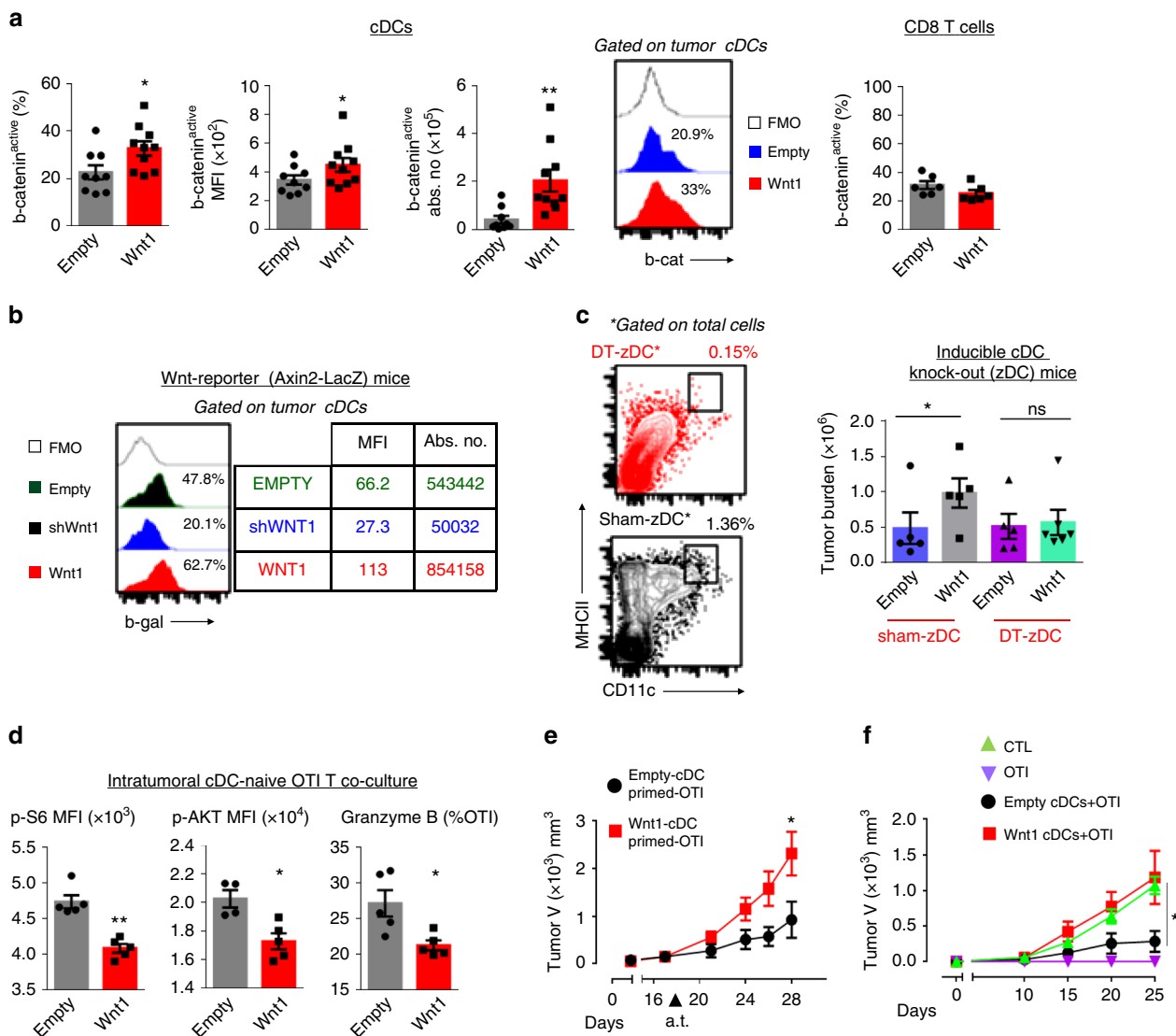

**Fig. 4** Wnt1 overexpressing tumors depend on DC signaling to evade T cells. Intratumoral cDCs were analyzed by FACS. **a** Percentages of b-catenin active cDCs, MFI of b-catenin in cDCs, absolute number of b-catenin active cDCs and representative histogram plots (Left). Percentages of b-catenin active CD8+ T cells (right). **b** Histograms showing b-galactosidase expression (%) by intratumoral cDCs of Axin2LacZ Wnt-reporter mice (Left) MFIs and absolute numbers of galactosidase expressing cells (right). **c** FACS plots depicting the depletion efficiency of intratumoral cDCs in zDC mice after DT injection (Left). Tumor burden (total absolute number of LLC cells) in sham vs. DT-treated (cDC-depleted) zDC mice (right). **d** Naïve OTI T cells were co-cultured with purified cDCs from Wnt1-overexpressing vs. control OVA-LLC tumors. TCR pathway activation (p-S6 and p-AKT) and cytotoxic molecule expression (granzyme B) were measured by FACS. **e** OTI T cells that had been primed by intratumoral cDCs as in **d** were adoptively transferred to mice bearing established subcutaneous OVA-LLC tumors. Graph depicts tumor growth. **f** Healthy mice were co-adoptively transferred with naïve OTI T cells and purified cDCs from Wnt1-overexpressing vs. control OVA-LLC tumors. After 5 days mice were transplanted with OVA-LLC tumors. Graph depicts tumor growth. **a-f** Cell numbers and profiles were assessed by FACS. Error bars represent mean with SEM. Data are representative or cumulative of at least two independent experiments with 4-9 mice per group. *$p < 0.05$; **$p < 0.01$, Mann–Whitney. Source data are provided as a Source Data file

Wnt1-cDC-primed OTI T cells were less therapeutic upon transfer in OVA-LLC tumor-bearing hosts (Fig. 4e). To recapitulate these findings in vivo, we co-administered Wnt1-cDCs (or control cDCs) and naïve OTI T cells in healthy mice, allowed a 5 day interval for in vivo cross-priming to occur and then challenged the mice with subcutaneous OVA-LLC cells. Recipient mice of OTI-T cells plus control cDCs were not completely rescued from tumor growth, as did OTI-T recipient mice, but tumors did grow at slow rates. Wnt1-cDC recipient mice developed tumors that were similar to negative controls (Fig. 5f). Taken together the decreased TCR stimulation, low granzyme B secretion and impaired cytotoxicity of in vitro

and in vivo primed T cells by intratumoral Wnt1-cDCs, these data collectively show that Wnt1 suppresses the ability of cDCs to cross-prime T cytotoxic cells against cancer-specific antigens.

Albeit Wnt1 is most frequently overexpressed and serves as a strong negative prognostic factor in LUAD, other canonical Wnts maybe also found up-regulated in lung tumors[19–23]. Among these is the prototype canonical Wnt ligand Wnt3a. In hepatocellular and colon tumors, T cells and myeloid-like cells produce Wnt3a, which inhibits T cell differentiation towards effector cells[42, 44]. In addition, blocking Wnt3a antibody, administered in vivo, increases expression of the activation

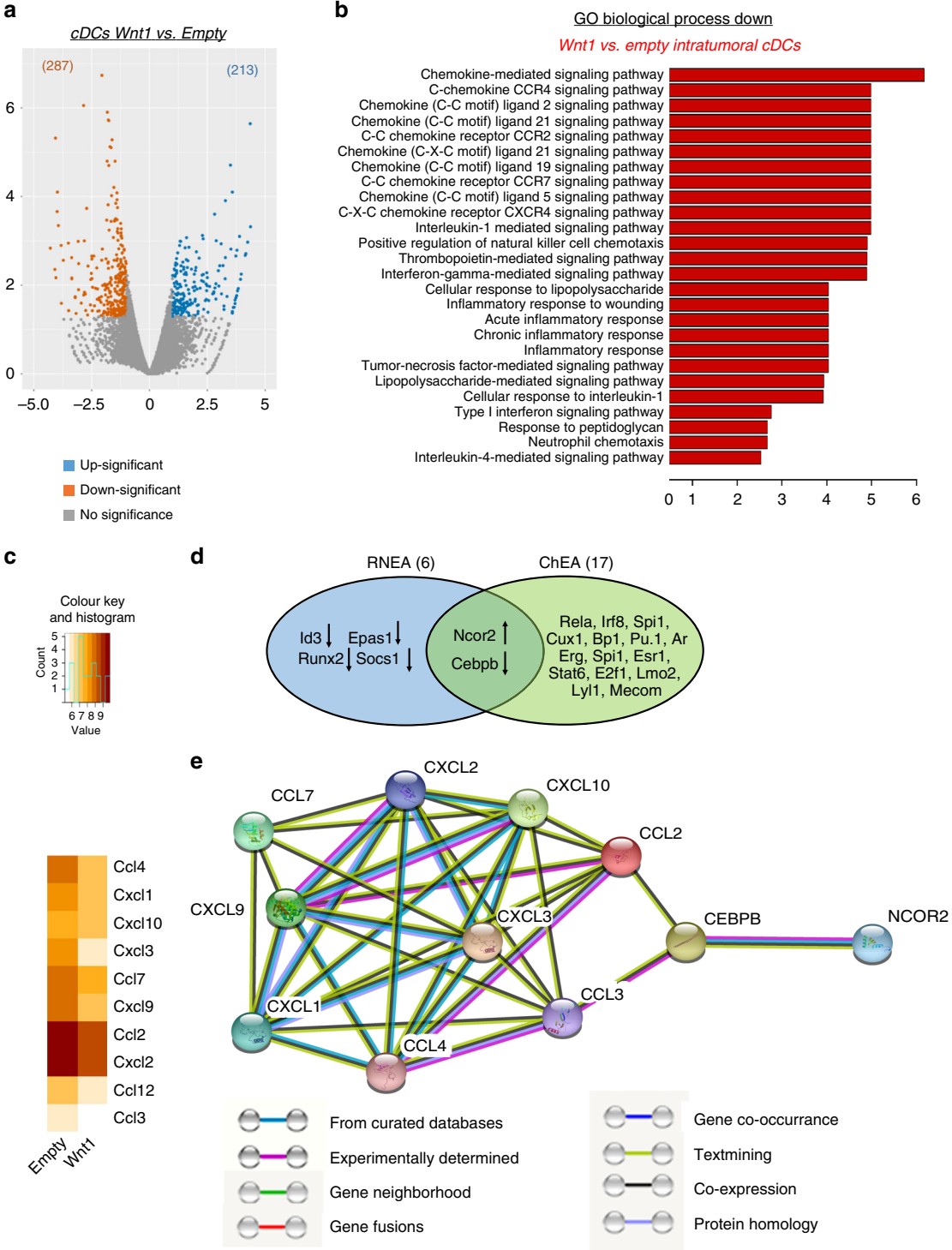

**Fig. 5** Wnt1 represses chemokine genes on intratumoral DCs. **a** Volcano plot of differentially expressed genes (DEGs) in cDCs purified from Wnt1-overexpressing (Wnt1) vs control tumors (Empty). **b** Functional enrichment analysis (GO Biological Processes) of downregulated genes in Wnt1 vs. Empty tumor cDCs. **c** Expression heatmap (log2 normalized read counts) of differentially expressed chemokines. **d** Venn diagram of the overlap of predicted regulators by RNEA and ChEA. **e** String database interactions for differentially expressed chemokines, Ncor2 and Cebpb (network indicates evidence with medium confidence for all active interaction sources). Intratumoral lung cDCs were pooled from 3–4 mice (n = 3 pooled samples per group). **d** Arrows indicate up or downregulation of the differentially expressed regulators in Wnt1 vs Empty tumor cDCs. Sequencing data are available with the accession code GSE123068

marker OX40L in tumor-infiltrating DCs[44]. We, therefore, explored whether inducing Wnt3a overexpression in LLC cells might have a similar impact to anti-tumor responses, as Wnt1 (Supplementary Figure 11). Wnt3a-overexpressing cells showed a significant growth advantage in vivo, compared to Empty cells.

However, T cells were not excluded from Wnt3a tumors. CD44 was relatively low in intratumoral T cells, which may be due to direct WNT3a-induced suppression[42]. Therefore, the immunological profile of Wnt3a overexpressing LUADs is independent of direct cDC signaling.

**Wnt1 represses chemokine genes on intratumoral cDCs**. Based on these observations, we next sought to comprehend why cDCs were rendered incompetent in the context of Wnt1 over-expression. Intratumoral cDCs that do not express the Wnt co-receptor LRP show enhanced cancer antigen uptake and lymph node migration[45]. Furthermore, lung specific Wnt1 over-expression decreases allergen uptake and DC migration to regional lymph nodes[46]. We, therefore, assessed the ability of cDCs to capture antigens and travel to mesothoracic lymph nodes by inoculating LLC cells expressing the fluorophore mCherry in the lungs of syngeneic mice. No differences were detected between Wnt1 overexpressing and control tumors (Supplementary Figure 12). The inhibitory mediators PD-L1 and IDO are reportedly increased in b-catenin-active cDCs[11, 13]. However, intratumoral cDCs of control LLC lung tumors were already uniformly expressing PD-L1 and IDO at very high levels by FACS analysis and no further increase could be detected upon Wnt1 overexpression (Supplementary Figure 12). Transcriptionally active b-catenin is also known to shift the cytokine balance towards immunosuppressive[11, 13, 34, 35, 45], so we used cytometric bead array analysis to quantify IL-10, IL-27, IL-12, and TNFa in culture supernatants of stimulated purified cDCs. No gross differences were again detected, besides a small increase in IL-27 upon lipopolysaccharide stimulation (Supplementary Figure 12). We also assessed chemokine receptor expression by FACS analysis and detected no differences between Wnt1 overexpressing and control tumors (Supplementary Figure 13). The aforementioned negative data led us to hypothesize that there was a previously unreported mechanism linking Wnt/b-catenin activation and cDC suppression.

RNAseq analysis of purified Wnt1-cDCs vs. Empty-cDCs revealed a number of differentially expressed genes, the majority of which were downregulated in Wnt1-cDCs (Fig. 5a). Functional enrichment analysis of downregulated genes showed a vast enrichment of chemokine signaling-related GO Biological Processes (Fig. 5b). A less pronounced decrease in MHCI was observed (FDR = 0.052). When examining chemokine ligands in the expression profile, 10 of them were found differentially expressed and all of them were found to be downregulated in Wnt1-cDCs (Fig. 5c). Downregulation of the 5 most prominently decreased genes was confirmed via qPCR, while MHCI down-regulation was confirmed by FACS, in distinct cohorts of mice (Supplementary Figure 14).

Having found decreased chemokine expression in cDCs infiltrating Wnt1 overexpressing tumors we sought to investigate its functional outcome on T cell priming. To this end we co-cultured intratumoral cDCs sorted from OVA-LLC tumors with naïve OTI T cells in the presence of chemokine blocking antibodies. Blocking chemokines suppressed CD44 expression and production of the effector molecules IFN-γ and granzyme B by OTI T cells, indicating a strong dependence of T cell activation on chemokine signaling (Supplementary Figure 15).

We next inspected the putative regulatory molecules that may play a role in the repression of the chemokine signaling. First, enriched regulators in downregulated genes were extracted with the use of RNEA tool[47]. With this approach 6 regulators were found enriched, i.e. Id3, Cebpb, Epas1, Ncor2, Runx2, and Socs1 (Fig. 5d). Aiming to further filter out this list of regulators and strengthen our results, the ChEA enrichment calculation was used with the downregulated chemokines as input. The overlap of these two approaches lead to Cebpb and Ncor2, which were down- and up-regulated respectively in Wnt1-cDCs (Fig. 5d). Interestingly, Cebpb is a transcription factor which promotes chemokine gene transcription[48–50], while Ncor2 is a co-repressor of Cebpb[51, 52], suggesting that Wnt1/b-catenin signaling might silence chemokine genes by up-regulating Ncor2 (Fig. 5e).

**Wnt1 RNA interference rescues cDCs and halts tumor growth**. Despite the importance of Wnt signaling in several cancers, Wnt inhibitors are not currently approved for use in the clinic largely due to their substantial toxicity[53]. Nanoparticle delivery systems have been extensively used in cancer therapy, as they are passively targeted to tumors through the enhanced permeability and retention effect[54]. In addition, they can be loaded with genome-engineering vectors[55], which makes them particularly suitable for the effective application of personalized medicine. We loaded siWnt1 RNA to DOPC liposomes and tested their therapeutic potential against Wnt1 overexpressing LUAD cells in vitro and in syngeneic orthotopic models. SiWnt1-DOPC decreased Wnt1-LLC cell proliferation rates in vitro (Fig. 6a). In vivo, Cy3-loaded control liposomes densely localized in LLC lung tumors (Fig. 6b) and siWnt1 liposomes efficiently decreased Wnt1 expression in LLC[mcherry] cells, but not in host's cells (Fig. 6c). We ruled out the possibility of off target effects of Wnt1 siRNA in DCs and mac-rophages, which can uptake liposomes, by assessing Wnt1 expression by FACS. Neither cell type expressed Wnt1 at the protein level (Supplemetary Figure 16). In accordance, our purified intratumoral cDCs did not show Wnt1 gene expression by RNA sequencing (GSE123068). SiWnt1 treatment was well tol-erated, further suggesting that Wnt1 was rather not necessary for adult murine tissue homeostasis. Importantly, in vivo RNA interference against Wnt1 rescued intratumoral cDCs from b-catenin activation (Fig. 6d). B-catenin activation has been pre-viously shown to decrease cDC responses to vaccination with a-DEC205-OVA[34]. SiWnt1-lodaded nanoparticles were effective as monotherapy and synergized remarkably to aDEC205-OVA plus adjuvant (poly I/C) (Fig. 6e). We also questioned whether siWnt1 might act therapeutically against wild-type LLC tumors. We did observe a less impressive, but still significant response (Supple-mentary Figure 17). Flt3L is an essential cytokine for the gen-eration of DCs[56]. Flt3L monotherapy had a small impact on LUAD growth, but worked very well in combination with siWnt1 (Fig. 6e).

Autochthonous murine tumors may more closely recapitulate human tumor–host interactions, so we validated Wnt1 as immunotherapeutic target in autochthonous lung adenocarcino-mas. Although Kras mutant genetically engineered mouse models are commonly used to test novel therapeutic targets, kras mutant lung adenocarcinoma cells express low Wnt1[2]. This was confirmed by our own preliminary experiments (Supplementary Figure 18). By contrast, Wnt1 was higher in urethane-induced lung tumors compared to healthy lungs (Fig. 6f). We, therefore, treated mice with established urethane-induced lung adenocarci-nomas with siWnt1-loaded vs. control nanoliposomes. siWnt1 treatment reduced tumor burden, accompanied by increased numbers of T cytotoxic cells and decreased b-catenin active cDCs (Fig. 6f).

Based on these findings, we hypothesized that blocking Wnt1 signaling in human LUADs might also rescue intratumoral cDCs from Wnt pathway activation. First, we analyzed the gene expression profile (RNAseq) of purified primary human LUAD cDCs focusing on Wnt-pathway target genes (Supplementary Table 2) (Supplementary Figure 19)[36, 57]. Among the two major subset of cDCs, i.e., CD1c− and CD1c+ cDCs, the CD1c− subset is more efficient in antigen cross-presentation[58]. Although Wnt-pathway target genes (Supplementary Table 3) did not differ between intratumoral and juxtatumoral CD1c+ cDCs, they were significantly upregulated in CD1c− cDCs (Fig. 7a). To sub-stantiate a specific role for human Wnt1 in b-catenin activation in intratumoral cDCs, we cultured dissociated primary lung adenocarcinomas in the presence or absence of siWnt1 RNA. Targeting Wnt1 decreased levels of active b-catenin in human cDCs - but not in T cells-, furthermore increasing T cytotoxic cell

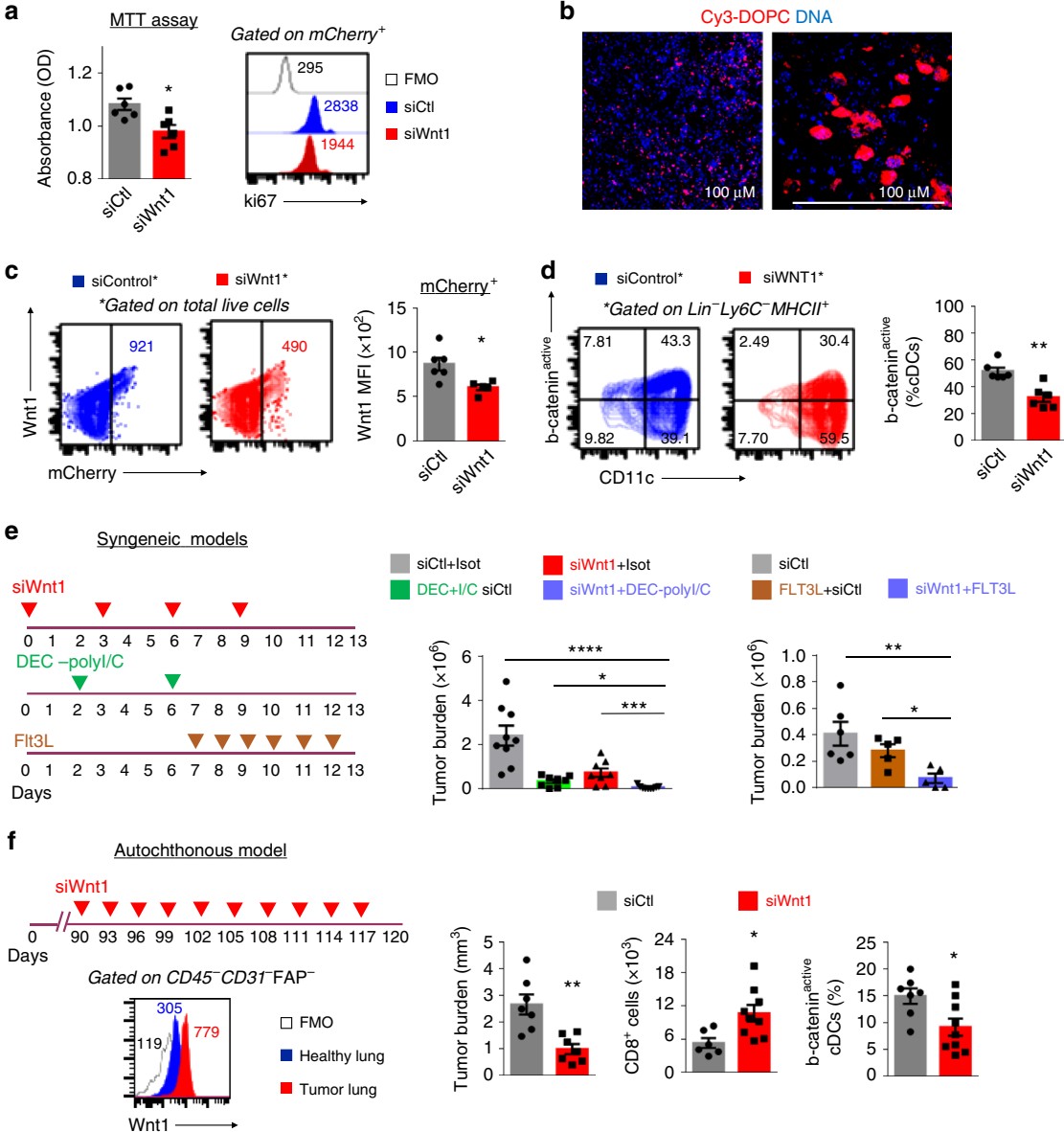

**Fig. 6** RNA interference against Wnt1 rescues cDCs from b-catenin activation and synergizes with DC-targeted therapies. **a** In vitro proliferation of Wnt1 overexpressing LLC cells in the presence of siWnt1 RNA nanoliposomes. MTT assay (left) and Ki67 expression (right). **b** Confocal photos showing lung tumor localization of intraperitoneal administered Cy3 loaded DOPC nanoliposomes. **c** siWnt1 or siControl RNA nanoparticles were administered to mice bearing Wnt1 overexpressing OVA-LLC lung tumors in vivo. FACS contour plots of Wnt1 expression by mCherry$^+$ LLC and mCherry$^-$ tumor stroma cells (Left). Cumulative data (right). **d** Representative FACS contour plots depict active b-catenin expression by cDCs (left). Cumulative data (right). **e** In vivo lung tumor growth. siWnt1 was given therapeutically either as single agent or in combination with a-DEC205-OVA plus polyI/C (left) or Flt3Ligand (right). **f** Immunotherapeutic efficacy of siWnt1-loaded nanoparticles in autochthonous urethane-induced lung adenocarcinomas. siWnt1 or siControl RNA nanoparticles were administered to mice bearing established urethane-induced lung tumors in vivo (left up). Representative FACS histogram plots of Wnt1 expression by CD45$^-$CD31$^-$FAP$^-$ cells (left bottom). Cumulative data showing tumor burden, numbers of lung T cytotoxic cells and b-catenin active cDCs (right). **c–f** Data are representative or cumulative of at least two independent experiments with 4–9 mice per group. Error bars represent mean with SEM. *$p < 0.05$; **$p < 0.01$, Mann–Whitney. Source data are provided as a Source Data file

membrane CD107, a degranulation marker (Fig. 7b). Taken together the in vivo immunotherapeutic efficacy of murine siWnt1 with the aforementioned in vitro effects of human siWnt1 on intratumoral cDCs and T cytotoxic cells, these data substantiate the immunotherapeutic value of Wnt1 in LUAD.

## Discussion

This study suggests that secretion of oncogenic Wnt1 in LUAD may act to suppress chemokine genes in cDCs and induces immunologically cold tumors. We observed strong in vivo

associations between Wnt1, T cell exclusion and biomarkers of immune tolerance in two distinct cohorts of LUAD patients. Mouse experiments showed that in contrast to what has been previously reported for melanoma-related b-catenin, Wnt1/b-catenin does not impact tumor cDC infiltration. It rather acts to silence chemokine genes in cDCs. Accordingly, in vivo RNA interference against Wnt1 not only impacted cancer cell autonomous proliferation but also rescued cDCs from b-catenin activation, leading to retardation of lung tumor growth. Additionally, siWnt1 RNA synergized in vivo with DC-targeted vaccination

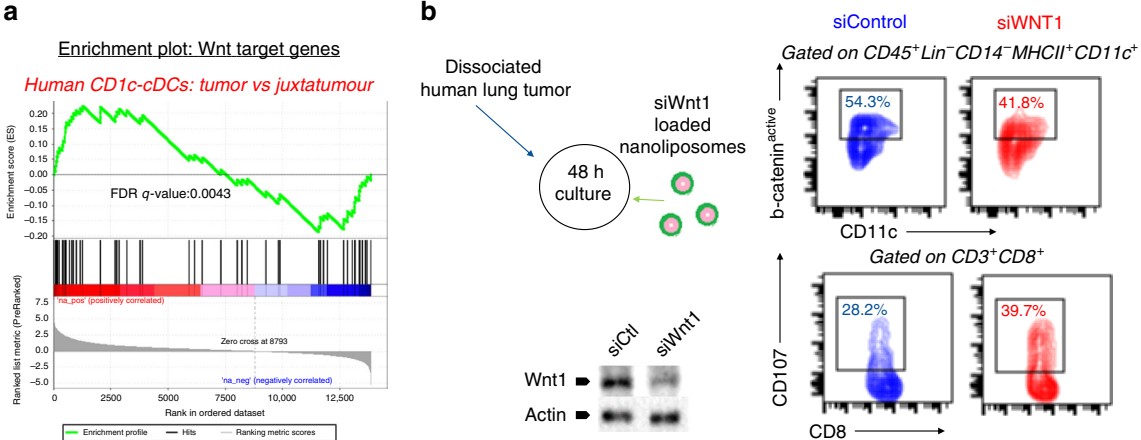

**Fig. 7** Wnt1 silencing attenuates Wnt pathway activation in human LUAD cDCs. **a** Human CD1−cDCs were sorted from primary lung adenocarcinomas and paired juxta-tumor healthy lung and analyzed by RNAseq. Gene Set Enrichment Analysis (GSEA) enrichment plot ($n = 4$ per group). **b** Primary human lung adenocarcinomas were dispersed and total cells cultured in the presence of Wnt1-silencing RNA (siWnt1) or control siRNA loaded nanoparticles (Up). Wnt1 silencing was confirmed by western blot (bottom). Representative histograms showing active b-catenin in cDCs (up) and extracellular CD107 in CD8+ T cells (bottom). Data are representative of 3 independent experiments. Source data are provided as a Source Data file. Sequencing data are available with the accession code GSE124199

and the DC development cytokine Flt3L. Finally, Wnt-target genes were up-regulated in human LUAD cDCs, while targeting human Wnt1 ex vivo rescued cDCs from b-catenin activation, increasing T cell cytotoxicity and opening avenues in LUAD therapy.

A growing series of activating mutations in Wnt pathway components have been reported in cancer throughout the years, delaying recognition of the importance of Wnt ligands as intercellular tumor signals. Considering that most types of cancer do not bear identifiable Wnt pathway mutations but still exhibit prominent intracellular Wnt pathway activation, intercellular signaling is likely to be the prevailing mode of function of Wnts in cancer[59]. In support, colon cancer cells with APC mutations depend on Wnt ligand signaling for sufficient b-catenin activation[60]. In LUAD a subpopulation of Wnt-secreting cancer cells form a Wnt-providing niche that enhances stemness of neighboring cancer cells[2]. The role of secreted Wnts on tumor–host interactions is also beginning to emerge. Upregulating the Wnt ligand secretion machinery in teratoma cells decreases T and B cell infiltration[61]. Melanoma-derived Wnt5a triggers b-catenin activation and metabolic reprogramming of cDCs[11, 53]. Although oncogenic Wnt1 is one of the strongest existing negative prognostic factors in human LUAD, so far any immune-related functions for this or any other LUAD-derived Wnts were unknown[19–23]. The data presented here clearly demonstrate that LUAD-derived Wnt1 is a paracrine suppressive signal for cDCs and thus highlight Wnt1 as therapeutic target.

An interesting question that arises from our findings is whether defective T cell priming by Wnt1-exposed cDCs occurs in lymph nodes or in the tumor microenvironment. By stark contrast to intratumoral cDCs, the b-catenin pathway was not activated in nodal cDCs of Wnt1 overexpressing lung tumors. We, therefore, speculate that reduced chemokine expression by intratumoral cDCs impacts effector T cell trafficking and priming at tumor sites.

Agents that universally block Wnt ligand secretion or inhibit the function of multiple Wnt(s) are currently being tried in the clinic, but there are safety concerns from disturbing elements that are necessary for healthy tissue homeostasis[3]. In addition, there are non-canonical Wnt(s) that antagonize b-catenin activation and can inhibit cancer progression[3, 62]. A strategy that selectively targets single cancer cell-derived Wnt(s) has not been adopted so

far, based on the concept that cells will use alternative ligands to transduce the signal if one specific ligand is blocked. Our studies refute this hypothesis. Silencing human and murine Wnt1 rescued DCs from b-catenin activation and restored anti-tumor immunity, suggesting that Wnt1 plays non-redundant roles in adaptive immune resistance of lung tumors.

There are strong in vivo associations between cDC and T cell numbers across most human cancers, including LUAD, which confirm the pivotal role of cDCs in T cell migration at lymphoid organs and peripheral tissues and their primary and secondary cross-priming[16]. Still, subsets of patients present with adequate numbers of cDCs but few T cells[63]. We show herein that one mechanism of T cell unresponsiveness in cDC-sufficient LUAD is driven via Wnt1 inhibition of CC and CXC motif chemokine genes in cDCs. cDCs, particularly cDCs1, are key cellular sources of T cell-attracting chemokines at tumor sites[10]. Chemokines not only attract T cells at sites of antigen presentation, but also enhance T cell priming by prolonging the contact duration between T cells and DCs[64, 65]. In line with this, blocking chemokines in cDC-T cell co-cultures in vitro sufficed to impair T cell priming. It could be postulated that the observed cDC defects are due to alterations in the composition of the cDCs. However, we observed no difference in numbers of cDCs1 and cDCs2 between Wnt1 overexpressing relatively to control tumors. Nevertheless, a differential effect of Wnts on DC subsets cannot be excluded and remains an interesting topic for future research.

Albeit downregulation of MHCI was expected to decrease the cross-presenting ability of cDCs, Wnt1-exposed cDCs efficiently cross-presented mCherry in vivo. One potential explanation is that mCherry cross-presentation does not recapitulate cross-presentation of other cancer antigens. On the same line, we did not observe a decrease in the expression of genes that regulate cross-presentation pathways. Markers of DC activation, such as co-stimulatory and co-inhibitory molecule expression and effector cytokine secretion, were also unaffected by Wnt1 overexpression. Even though we did not detect defects in these crucial immune functions of cDCs, DC chemokine deficiency alone, may well explain the T cell exclusion from tumor sites and unresponsiveness against tumor antigens.

Interestingly, Wnt1 downregulated the transcription factor Cebpb in intratumoral cDCs. Cebpb has several binding sites for CC, CXC chemokine genes at their promoters and enhances

chemokine gene translation[48–50]. In melanoma cells, b-catenin translocation to the nucleus upregulates ATF3, which blocks the promoter of CCL4[4]. A plausible scenario would then be that active b-catenin upregulates expression of negative regulator(s) of CEBPB to repress chemokine gene transcription. Although Wnt1 overexpression did not affect ATF3 in cDCs, it up-regulated a known Cebpb co-repressor, Ncor2[51, 52]. We are working to further define the Wnt1/b-catenin-driven molecular mechanisms that inhibit chemokine production by cDCs, hoping to reveal novel insights into the complex regulation of cross-priming and open new avenues for optimizing cancer immunotherapies.

Most importantly, our current studies provide a specific framework for cancer immunotherapy through Wnt1 inhibition, acting to unleash DC immunity, coupled to cDC expansion with Flt3L and/or selective targeting of tumor antigens on cDCs via aDEC205. What makes Wnt1 a particularly promising immunotherapeutic target in LUAD is that: (i) it is frequently found upregulated in LUAD patients and is one of the strongest negative prognostic factors[19, 20, 22, 23, 30, 66], (ii) its expression is higher in tumors than in healthy tissues[19, 22, 23, 66] and iii) human LUAD cells use autocrine Wnt signaling to increase their proliferative and stemness potential[2]. Therefore, our targeted breakthrough is expected to have vast therapeutic implications for human LUAD, acting concomitantly at both arms of the cancer-immune bionetwork.

## Methods

**Patients**. Archived formalin-fixed, paraffin-embedded (FFPE) lung adenocarcinomas (LUADs), and fresh LUADs (plus juxta-tumor normal lung tissue when required) were provided by Sotiria and Evangelismos General Hospitals, with informed consent and under authorization of the ethical committees of the hospitals (Supplementary Tables 1, 2). LUADs were clinically scored and staged according to the International Union against Cancer (UICC) TNM staging system. Fresh samples were cut in two blocks and either FFPE or pushed through strainers to obtain single cell-suspensions for FACS sorting/ex vivo assays. FFPE tissue blocks were sectioned at 5 µm and stained with hematoxylin and eosin (H&E) to identify tumor.

**Mice**. The following mouse strains were used: the C57Bl/6 (Jackson Laboratories, ID #000664) and FVB/NJ (Jackson Laboratories, ID #001800) wild type mice. For T cell responses against ovalbumin-expressing cancer cells, OTI mice (strain 002929003831) that express a transgenic TCR designed to recognize ovalbumin residues 257–264 in the context of H2Kb (kind gift from Andreakos Lab, BRFAA). For cDC deletion studies, the B6(Cg)-Zbtb46^tm1(HBEGF)Mnz/J (zDC^DTR) mice that encode the human diphtheria toxin receptor specifically in cDCs (Jackson Laboratories, strain 019506). For adoptive cell transfer studies, the B6.SJL-Ptprc^a mice that express the CD45.1 (Ly5.1 PTP) alloantigen (Jackson Laboratories, strain 002014). For adoptive transfer of OTI T cells, OTI TCR transgenic mice were crossed to B6.SJL-Ptprc^a mice. For Wnt/b-catenin pathway reporter studies, the B6N.129P2-Axin2^tm1Wbm/J mice were used (kindly provided from Jurgen Behrens Lab, University of Erlangen). For Kras-driven adenocarcinomas the B6.129S4-Kras^tm4Tyj/J (Jackson Laboratories, strain 008179). Gender-matched 8–12-week old mice were used for all studies. All mice were housed under standard special pathogen-free conditions at BSRC Alexander Fleming, except: the B6N.129P2-Axin2^tm1Wbm/J mice, that were housed at the Animal Model Research Unit of Evagelismos Hospital and the FVB/NJ and B6.129S4-Kras^tm4Tyj/J mice, that were housed at the University of Patras Center for Animal Models of Disease. All animal procedures were approved by the Veterinary Administration Bureau, Prefecture of Athens, Greece under compliance to the national law and the EU Directives and performed in accordance with the guidance of the Institutional Animal Care and Use Committee of BSRC Al. Fleming.

**Cell lines**. The Lewis Lung Carcinoma cell line (LLC) was obtained from American Type Collection Cultures (Manassas, VA). The 43 ATCC-TIB-210 hybridoma cell line, expressing the monoclonal antibody against Lyt-2.2, was purchased from ATCC (Rockville, MD). The following cell lines were kind gifts: the C57Bl/6-derived urethane-induced lung adenocarcinoma (CULA cells)[67] and FVB-derived urethane-induced lung adenocarcinoma (FULA cells)[31] (Stathopoulos Lab, Department of Medicine, University of Patras), the mouse colon-26, AB1, AE17, MC-38 (Kalomenidis Lab, Department of Intensive Care Medicine, University of Athens), the mouse CMT-39 cell line (Kontoyiannis Lab, BSRC Al. Fleming), the platinum-E (Capetanaki Lab, BRFAA) and the HEK lenti-X 293T cells (Fousteri Lab, BSRC Al. Fleming). All the cells were tested negative for the presence of

mycoplasma contamination using a PCR-based technology. Cell lines were not authenticated.

**Reagents and resources**. Reagents and resources are included in Supplementary Table 4.

**Primary human cultures**. Primary LUADs were pushed through 70 µm strainers (Corning) and cultured for 48 h in 96-U bottom plates in RPMI−1640 (Gibco) supplemented with 10% human serum, 1% L-glutamine, 1% penicillin and streptomycin, in the presence of control or siWnt1-loaded DOPC nanoliposomes (5ug per $4 \times 10^6$ cells).

**Human cDC sorting and processing**. Primary LUADs and juxta-tumor lung tissues were pushed through 40µm strainers (Corning). CD45+ cells were magnetically enriched using CD45 Microbeads (Miltenyi Biotech), stained as described in Supplementary Figure 19 and sorted in RL buffer (NorgenBioteck). RNA purification was performed with Norgen kit (NorgenBioteck) and cDNA synthesis with SMARter kit (Clontech).

**Immunohistochemistry**. Immunohistochemical stainings were performed on 5 µm formalin-fixed, paraffin-embedded sections or 10 µm frozen sections. Incubation with the primary antibodies was tested in various ways to obtain low background positivity and high signal to noise ratio. Negative control experiments for non-specific binding were performed by replacing the primary antibody by non-specific IgG of the same species as the primary antibody or the pre-incubation solution. Briefly, for fixed sections, antigen retrieval was performed using the EnVision™ FLEX Target Retrieval Solution, High pH(DAKO) for 30 min. Endogenous peroxidase was blocked with 0.3% $H_2O_2$ for 15 min at room temperature (RT), followed by incubation with universal blocking buffer (DAKO) containing 1%BSA, 0.5% Triton X-100, 0.05% sodium azide and 0.01 M PBS at pH 7.2–7.4. Primary antibodies were diluted in Antibody Diluent (DAKO). The following staining conditions were applied: for anti-human CD8 detection 1:100, 30 min, RT (Dako, C8/144B), for anti-human granzyme B detection 1:100, 30 min, RT (Thermo, PA1-37799), for anti-mouse CD8 detection 1:50, overnight, 4 °C (Santa Cruz, sc-7188). The EnVision™ FLEX detection system (DAKO) was applied for 30 min at RT. Immunoreaction was visualized using EnVision™ FLEX DAB+ Chromogen for 1 min at RT. Sections were counterstained with EnVision™ FLEX Hematoxylin, dehydrated, and mounted. Sections were coded and counted by a single blinded observer (D.K.) with Eclipse E800 microscope (Nikon).

**FFPE RNA extraction and qPCR**. For the RNA extraction from formalin-fixed, paraffin-embedded human tissue, the RecoverAll Total Nucleic Acid Isolation Kit (Invitrogen) was used according to manufacturer's recommendations. RT-PCRs where performed with primers sets purchased from Sigma-Aldrich (H_WNT1_1, H_WNT10A_1, H_WNT10B_1, H_WNT2_1) and human U6 snRNA Fw 5'-CTC GCTTCGGCAGCACA-3' and Rv 5'-AACGCTTCACGAATTTGCGT-3'.

**Syngeneic tumor models**. For the orthotopic lung cancer model, mice were anesthetized via i.p. injection of xylazine and ketamine. Cancer cells ($2 \times 10^5$) resuspended into 50 ul DMEM and enriched with 20% extracellular matrix (Matrigel, BD Biosciences) were intrapleurally injected to the lung parenchyma of mice using a 29G needle (BD Biosciences). For the subcutaneous cancer model, mice were anesthetized as above and injected subcutaneously on the left shaved flank with $5 \times 10^5$ tumor cells (LLC) suspended in 100 µl DMEM. Flank tumor masses were measured by assessing length and width using a digital caliper. Tumor volume was calculated as (length × width$^2$)/2. For the metastatic model, mice were injected intravenously via the tail vein with $2 \times 10^5$ LLC cells in 200 µl DMEM.

**Autochthonous cancer models**. Chemical-induced lung adenocarcinoma was induced in FVB mice by a single intraperitoneal exposure to 1 g/kg urethane[68]. KRASG12D-driven LADC was induced via intratracheal injections of $5 \times 10^8$ plaque-forming units (PFU) adenovirus type 5 encoding CRE recombinase (Ad-Cre) to LSL.KRASG12D mice on the C57BL/6 background[69]. Mice were sacrificed and lungs were harvested at 120 days post-urethane or post-Ad-Cre.

**In vivo T cytotoxic cell depletion**. Mice were given i.p. 150 µg of the depletion antibody 2 days prior to tumor inoculation, and were subsequently injected three times per week with 150 µg of the depleting antibody.

**In vivo T cell proliferation**. The thymidine analog 5−bromo-2′-deoxyuridine (BrdU, Roche) (10 mg/ml, 200 µl/20 g mouse) was injected i.p. into tumor-bearing hosts. The day after mice were sacrificed and the number of the T cells that had incorporated BrdU, was determined by FACS. Alternatively, FACS-sorted cells T cells were CFSE-stained (10 uM, 15 min, 37 °C) (C34554, Molecular probes) prior their adoptive transfer to tumor-bearing hosts. T cell proliferation was assessed after 3 days by CFSE dilution using FACS.

**Adoptive cell transfers**. For T cell transfer, congeneic CD45.1$^+$ T cells were FACS-sorted from splenic or tumor single cell suspensions or cDC-T cell co-cultures as Ter119$^-$MHCII$^-$CD105$^-$CD11b$^-$CD11c$^-$CD19$^-$B220$^-$NK1.1$^-$. Each mouse received $1.5 \times 10^6$ splenic or $4 \times 10^5$ intratumoral T cells or $8 \times 10^4$ cDC-primed T cells in 200 ul PBS via the i.p. route. For cDC-T cell co-transfer, CD45.1$^+$ OTI T cells FACS-sorted from splenic single cell suspensions and cDCs FACS-sorted from tumors as described in Supplementary Figure 7 were transferred to mice. Each mouse received $7.5 \times 10^5$ intratumoral T cells and $6 \times 10^4$ cDCs cells in 200 ul PBS via the i.p. route.

**siWnt1 immunotherapy**. siWnt1-loaded or control nanoliposomes were given i.p. route in mice at a dose of 150 ug/kg/mouse in a volume of 200 uL PBS/mouse as shown in Fig. 6 and Supplemetary Figure 17.

**DC-target immunotherapies**. Mice were primed twice i.p. with 10 ug aDEC205$^-$OVA in the presence of 100 ug poly I/C or isotypic control. Alternatively, 10 ug/mouse human recombinant Flt3 protein (Flt3L) (gift from Panagiotis Tsapogas, University of Basel) were administered daily i.p. as shown in Fig. 6e.

**Murine tumor cDC sorting and processing**. Murine tumors were pushed through 40μm strainers (Corning). CD45$^+$ cells were magnetically enriched using CD45 Microbeads (Miltenyi Biotech). Enriched cells were stained as described in Supplementary Figure 7 and sorted in culture medium or in RLT buffer for RNA extraction (RNeasy micro kit, and DNase set Qiagen). Spleen cell were depleted for B and T cells using Dynabeads (Invitrogen), anti-B220 (553084, BD Biosciences) and anti-CD3 (100202, Biolegend), then stained and sorted as above. MMLV (Thermo) was used for reverse transcription and SYBR Green (Thermo) for qPCR (CFX96 Touch™ Real-Time PCR Detection System, Bio-Rad). Alternatively, one step RT-PCR was performed from purified RNA using QuantiTect SYBR Green (Qiagen). RT-PCRs where performed with primers purchased from Sigma-Aldrich (M_CCL2_1, M_CCL4_1, M_CCL7_1, M_CXCL3_1, M_CXCL9_1, M_CEBPB_1, M_NCOR2_1) and mouse B2M, *Fw:* 5'-TTCTGGTGCTTGTCTCACTGA-3' and *Rv:* 5'-CAGTATGTTCGGCTTCCCATTC-3'.

**Primary murine cell cultures**. All primary murine cells were cultured in RPMI-1640 (Gibco) supplemented with 10% fetal bovine serum, 1% L-glutamine, 1% penicillin and streptomycin. For cytokine measurement FACS sorted cDCs were stimulated with LPS 1 ug/ml (Sigma-Aldrich) or poly I/C 25 ug/ml (Sigma-Aldrich) (15.000 cells/200 ul RPMI/well). Cytokines IL-27, IL-12, IL-10, TNFa were detected in the culture supernatants using a customized multiplex bead-based immunoassay (Mouse inflammation Panel, LEGENDplex, Biolegend) according to the manufacturer's recommendations. Data were recorded by a FACSCANTO II and analyzed using the "LEGENDplex" Data Analysis software 7.0. For b-catenin activation sorted cDCs were exposed to 100 ng/ml rWnt1 (9765-WN-010, R&D) or culture supernatants from Wnt1 overexpressing or control cells lines LLC, CULA and FULA. FACS sorted cDCs were co-cultured with purified splenic OTI T cells at a ratio of 1:5. In selected experiments blocking antibodies against CCL3 (0.2 ug/ml), CCL4 (3 ug/ml), CXCL9 (10 ug/ml), CXCL10 (20 ug/ml) and CXCL11 (10 ug/ml) (Thermo) were added. Chemokines were detected in culture supernatants using customized multiplex bead-based immunoassay (Mouse Proinflammatory Chemokine Panel, LEGENDplex, Biolegend) according to manufacturer's recommendations. IL-2 was added after day 3 (10 ug/ml, Promega). T cells were collected on day 6 and either analyzed by FACS or adoptively transferred in tumor-bearing mice. For IFN-γ detection single-cell suspensions from murine lung tumors were stimulated with 1000 ng/ml ionomycin and 500 ng/ml phorbol 12-myristate 13-acetate (PMA) for 4 h in the presence of brefeldin A (Biolegend), followed by FACS staining and analysis. For in vitro cytotoxicity assays, total intratumoral T cells were FACS-sorted from tumor single cell suspensions as Ter119$^-$MHCII$^-$ CD105$^-$CD11b$^-$CD11c$^-$CD19$^-$B220$^-$NK1.1$^-$. Intratumoral CD45.1$^+$ OTI T cells were FACS-sorted based on CD45.1 staining. T cells were co-cultured with CFSE$^{high}$-LLC-OVA (CFSE, 5 μM) and CFSE$^{low}$-LLC (CFSE, 0.5 μM) at 60:1:1 (total T: CFSE$^{high}$-LLC-OVA: CFSE$^{low}$-LLC) or 12:1:1 (OTI T: CFSE$^{high}$-LLC-OVA: CFSE$^{low}$-LLC) ratio, centrifuged in V-bottom plates (1000 rpms, 10 min) and incubated for 4 h.

**Inducible cDC-knock out model**. To generate zDC$^{DTR}$-to-C57BL/6 bone marrow chimeras, C57BL/6 mice were irradiated using a cesium source (γ-irradiation), at two doses of 610 rad each, with a 3 h interval in-between. To obtain donor bone marrow from zDC$^{DTR}$ mice, femurs and tibiae from 8-week-old donors were harvested and the bone marrow was flushed out. 24 h after irradiation, $10^6$ HBSS-suspended (Gibco) total bone marrow cells were injected into mice i.v. Mice were kept on antibiotics for 2 weeks and assessed 3 months after transplantation. For depletion of cDCs, diphtheria toxin was administered i.p. one day pre-tumor engraftment (20 ng/g body weight) and was given every other day (4 ng/g body weight).

**Plasmid construction**. mCherry FP from pMSCV-IRES-mCherry FP vector (gift from Vignali Dario, University of Pittsburgh) was cloned into MIGR1-OVA-IRES-eGFP vector (gift from Zehn Dietmar, Swiss Vaccine Research Inst.) using NcoI and SalI restriction sites (NEB) to generate MIGR1-OVA-IRES-mCherry FP retroviral vector. pLNCX (empty vector) and pLNC-WNT1 plasmids (gifts from Jan Kitajewski, Columbia Uni Medical Center) were used for retroviral transductions of LLC cells. To generate Wnt1 overexpressing lentiviral vectors Wnt1 from pLNC-WNT1 plasmid was cloned into pHIV-Zsgreen plasmid (gift from Bryan E. Welm, Health Uni of Utah) as an XbaI fragment. pHIV-WNT1 and empty vector were used for lentiviral transductions of FULA and CULA cells. pGFP-C-shWNT1 vector and pGFP-C-scr vector control were bought from ORIGENE.

**Cancer cell line cultures**. Cells lines were maintained according to their recommendations in culture media containing DMEM (Gibco), 10% heat-inactivated fetal bovine serum (Biochrom), 1% L-glutamine, 1% penicillin and streptomycin (Gibco). For measurement of cell proliferation, MTT (5 mg/ml in PBS) was dissolved to 0.5 mg/mL final concentration in medium and added to each well of a 96-well plate seeded from the previous day with $10^4$ cells. After 4 h, the medium was replaced with 100 μL DMSO and 540 nm absorbance was measured (Optimax microplate reader). For selected experiments, MTT assays were performed in the presence siWnt1$^-$-loaded or control nanoparticles (1 ug per $4 \times 10^6$ cells).

**Transduction of cancer cell lines**. Platinum-E packaging cells were transfected with retroviral plasmids MIGR1-OVA-IRES-mCherry FP, LNCX or LNC-WNT1 using PEI (Polysciences). In brief, a total of 7 ug DNA was added to 500 μl OptiMEM medium (Gibco) and incubated for 5 min at RT. In parallel, 21 ug PEI were added to 500 μl OptiMEM medium and incubated for 5 min at RT. Dilutions of DNA and PEI were mixed and incubated for 15 min at RT and subsequently added to 10 cm plate containing confluent Platinum E cells. The cells were incubated for 16 h, after which the medium was replaced. Supernatants containing retroviruses were collected 48, 60, and 72 h after transfection and filtered (0.45 μm pore size, 83.1826, Sarstedt). Fresh retroviral supernatants were used for the transduction of LLC cells. LLC cells were transduced three times, with intervals of 8 h with virus supernatant containing 4 ug/ml polybrene (Sigma-Aldrich). 24 h after the first transduction virus supernatant was removed and replaced with fresh medium. 48 h after transduction cells were sorted for mCherry (BD FACSAria III, BD Biosciences) and expanded or selected for G418 resistance (400ug/ml). HEK lenti-X 293T cells were transfected with lentiviral plasmids pGFP-C-shWNT1 and scramble or pHIV-Wnt1 plasmid and empty vector using PEI as above. LLC, FULA and CULA cells were transduced three times with virus supernatant containing 8 ug/ml polybrene. 48 h after transduction cells were sorted for GFP and expanded or selected for puromycin resistance (1.5 ug/ml). LLC cells were also transduced with commercially available Wnt3a lentiviral particles (Wnt3a-RFP) or control (CMV-RFP), as instructed by manufacturer (Gen Target Inc), to generate Wnt3a over-expressing LLC cells and sorted for RFP. Cells were periodically checked for mCherry, RFP or GFP expression under fluorescent microscope (Axio Vert A.1, Zeiss).

**FACS analysis and sorting**. For FACS analysis and FACS sorting single-cell suspensions were resuspended in FACS buffer (PBS, 2% FBS, 1.5 mM EDTA). Cells were stained at 4 °C for 30 min with fluorescent-conjugated antibodies or primary antibodies, followed by fluorescent-conjugated secondary antibodies. The antibodies' dilutions applied were according to manufacturers' instructions. For detection of intracellular antibodies, phosphorylated proteins or transcription factors, cells were fixed and permeabilized using the Fixation & Permeabilization Buffer Set (eBioscience) according to manufacturers' instructions. Intranuclear BrdU detection was performed according to standardized protocols. For calculation of absolute numbers of tumor cells (burden) or immune cells counting beads (123Count eBeads, Thermo Fischer Scientific) were used. Briefly, a known volume of counting beads was added to the same known volume of stained cells. The beads were counted along with cells. The absolute count of the cell population (A) was obtained by dividing the number of positive cell events (X) by the number of bead events (Y) and then multiplying by the bead concentration (N/V, where N = number of beads per test and V = test volume). A = X/Y × N/V. FACS analysis or sorting was performed using FACSCANTO II (BD Biosciences) or BD FACSARIA III (BD Biosciences) and data were analyzed using Flowjo (Tree star Inc) or DIVA software (BD Biosciences). All gating strategies are shown in Supplementary Figures 7, 16, 19, 20.

**Quantification of tumor burden in autochthonous cancer models**. For chemical-induced LUAD, lungs and lung tumors were inspected macroscopically under a Stemi DV4 stereoscope equipped with a micrometric scale incorporated into one eyepiece and an AxiocamERc 5s camera (Zeiss, Jena, Germany) in trans-illumination mode, allowing for visualization of both superficial and deeply-located lung tumors. Individual tumor volume was calculated as $\pi \delta^3/6$. Total lung tumor burden per mouse was calculated as the sum of individual volumes of all tumors found in a given mouse lung.

**Immunofluorescence for visualization of Cy3-loaded nanoliposomes**. Lung tumor-bearing mice were treated i.p. with Cy3-loaded liposomes at a dose of 150 ug/kg/mouse 2 days prior to euthanasia. On day 14 lungs were excised, fixed in 4%

PFA for 1 h, followed by diving in 30% sucrose and freezing in OCT compound. 10μm cryosections were counterstained with DAPI. Images were acquired TCS SP8X White Light Laser confocal system (Leica) and analyzed with LAS X software.

**Immunoblot analysis**. Cells were lysed in RIPA buffer (50 mM Tris HCl, pH 8.0, 150 mM NaCl, 1% NP-40, 0.5% sodium deoxycholate, 0.1% SDS), containing protease inhibitors (Roche) and phosphatase inhibitors (Sigma-Aldrich). Western blotting was performed using standard protocols and according to manufacturer's recommendations. Antibodies Wnt1 (Abcam, ab15251), non-phospho b-catenin (Cell Signaling, 8814), Wnt3a (Abcam, ab19925) and actin (Santa Cruz, sc-1615) were used. Secondary antibodies conjugated with HRP were purchased by Vector Laboratories. Signal development was performed using the Luminata™ Crescendo Western HRP Substrate (Millipore) and signal acquisition was achieved using the ChemiDoc XRS + and Image Lab software (Bio-Rad). Uncropped plots are included in the source data file.

**ELISA**. Culture supernatants were obtained from Wnt1 overexpressing and control (Empty) tumor tissue fragments of lung tumors by incubating small fragments of tissue in complete RPMI for 16 h (100 mg/100 mL of medium). Wnt1 concentration in culture supernatants was measured using Wnt1 ELISA Kit (LS-F17163-1, LifeSpan BioSciences) according to manufacturer's instructions.

**Antibodies production**. Mice were depleted of CD8$^+$ T cells by i.p. injection of CD8-depleting antibody harvested from the hybridoma cell line 2.43, ATCC TIB 210. Briefly, exponentially growing cells ($2 \times 10^6$/ml) were seeded in serum free media complemented with 2.5% ultra-low IgG serum (Gibco), for 4 days. Culture supernatant was filtered (0.2 μm pore size, Sarstedt) and concentrated by ammonium sulfate precipitation by standard protocol (ACROS). Melon Gel IgG Spin Purification Kit (Thermo Fischer Scientific) was used for the antibody purification. Antibody purity was validated with SDS-PAGE and its concentration was determined by Nanodrop. Vectors encoding heavy and kappa chain from anti-DECOVA as well as isotype control antibody (provided by Michel C. Nussenzweig, The Rockefeller University) were used to transiently transfect in 293T cells using PEI in media complemented with 1% ultra-low IgG serum. The fusion antibodies were purified as described above.

**Manufacturing of siRNA-loaded nanoliposomes**. Human and mouse siWnt1, siRNA negative controls and fluorescent siRNA were purchased from Sigma-Aldrich (see key resources table). DOPC (Avanti Polar Lipids) was mixed in ratio 10:1 DOPC:siRNA in excess tertiary butanol and Tween-20 (in ratio 20:1, Sigma-Aldrich). The mixture was vortexed, frozen in an acetone/dry ice bath, lyophilized overnight and kept at 4 ºC until reconstitution in PBS.

**Statistical analysis**. We performed Pearson's correlation, *t*-test and Mann–Whitney *U*-tests with Prism software. We considered two-sided or one-sided *p*-values of less than 0.05 significant.

**Human cDC RNA-seq**. For the human DCs RNA-seq processing, the sequencing fastq files were first checked for sequencing quality using FastQC following the software recommendations. Then the reads were aligned using the software TopHat v2.0.6 with the genome reference version hg19 (GRCh37) and counted with HTSeq-count. All the following steps and analysis were performed using the software R (version 2.3.2). The reads were filtered excluding all the genes with less than 5 counts in more than 25% of the samples. The normalizing method used was the RUVr. Finally, to explore the data, PCA was performed using the 500 most variant genes selected using the inner inter-quartile range method of the EMA R package. Differential expression analysis (DEA) was performed comparing the juxta-tumor and tumor from the same donor (paired) for 13863 genes with edgeR package of R. The p-values were corrected by FDR. Enrichment analysis was calculated using the pre-ranked option of the software Gene Set Enrichment Analysis (GSEA). For the enrichment score a total list of 130 genes was used suggested to be Wnt target genes. Spearman correlations were calculated between gene expression levels and Wnt protein levels in the corresponding tumors of the patients. The correlations coefficients and corrected p-values were represented using R package Corrplot.

**Murine cDC RNA-seq**. The read counts table was created using the Bioconductor package GenomicRanges. The gene counts table was normalized using the Bioconductor package DESeq after removing genes that had zero counts over all the RNA-Seq samples. Differential expression analysis was performed using the Bioconductor package DESeq. This part of the analysis was performed through the Bioconductor package metaseqr. Volcano plots were generated in R with the use of ggplot2 package. The venn diagram was created with InteractiVenn. Functional enrichment analysis was performed with enrichr online tool focusing on the top enriched Gene Ontology Biological Processes. Prediction of enriched regulators was performed with RNEA (p value threshold 0.05) and enrichr focusing on the ChEA predictions with the downregulated chemokines as input.

**TCGA**. The RNA-Seq data were downloaded from TCGA (The Cancer Genome Atlas) using the Broad's Institure Firehose website. Spearman correlation was calculated between Wnt ligands and a selection of immunosuppressive genes. For each cancer type, the mean correlation values of Wnt ligands to immunosuppressive genes were ranked and all the ranks are visualized in a boxplot with R. Wnt1 correlation to CD8a and CD8b genes was calculated in tumor and normal tissues from TCGA, using the GEPIA tool. The distribution of the Spearman correlation values was examined with the calculation of z-scores across a series of different datasets. GTEx and TCGA (excluding the separately examined cancer types) were included as controls. Heatmaps were designed in R with heatmap.2 function from gplots package.

**Wnt ligands in LLC cells/tumors**. For the GSE58188 and GSE36568 datasets normalized signal intensity values were retrieved from the respective Series Matrix File. Heatmaps of the aforementioned values of Wnt ligands were generated in R using heatmap2 function from gplots package.

**Reporting Summary**. Further information on experimental design is available in the Nature Research Reporting Summary linked to this article.

## Data availability

Sequencing data that support the findings of this study are available in Gene Expression Omnibus (GEO) genomics data repository with the accession codes GSE124199 and GSE123068. RNA-Seq data underlying Fig. 1 and Supplementary Fig 1 were downloaded from TCGA [https://cancergenome.nih.gov/]. The source data underlying Figs. 1, 2, 3, 4, 6 and 7 and Supplementary Figs 3, 4, 6, 7, 9, 10, 11, 12, 13, 14, 15 and 16 are provided as a Source Data file. All the other data supporting the findings of this study are available within the article and its supplementary information files and from the corresponding author upon reasonable request. A reporting summary for this article is available as a Supplementary Information Files.

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

## Acknowledgements

We acknowledge Z. Dietmar, V. Dario, J. Kitajewski, B.E. Welm for plasmids, J. Behrens and E. Andreakos for mouse strains, P. Tsapogas for Flt3L, D. Kontoyiannis, Y. Capetanaki and M. Fousteri for cell lines, the NIH Tetramer Core Facility for H-2k(b) SIINFEKL, A. Apostolidou, V. Koliaraki, S. Grammenoudi, P. Michea, M. Katsa and C. Tzaferis for technical assistance. We thank the InfrafrontierGR Infrastructure for mouse facilities (co-funded by GR/EU, NSRF 2014–2020, ERDF, MIS 5002135). The project was supported by a Hellenic Thoracic Society Grant and a Stavros Niarchos Foundation grant to M.T., FP7 ERC MCs-inTEST (GA 340217) and the "Research program for Excellence IKY/Siemens" to G.K. and project MIS 5002562 (NSRF 2014–2020, ERDF, co-funded by GR/EU).

## Author contributions

Designing research study, M.Tsoum.; Conducting experiments, D.K., I. G., G.N. and M. Tsoum.; Analyzing data, M.Tsoum., D.K., P.C., Y.A.G., G.T.S. and A.R.A.; Providing reagents, G. Kol., V.S., G. Kaz., S.Z., I.K., M. Tsik., G.T.S. and K.P.; Writing, Review & Editing the manuscript, M. Tsoum., D.K., I.K. and G. Kol.; Funding Acquisition, M. Tsoum. and G. Kol.

## Additional information

**Competing interests:** The authors declare no competing interests.

