## [Peer Review File · Nature Communications]

Reviewers' Comments:

Reviewer #2:

Remarks to the Author:

Kerdidani et al. are reporting here on a novel way of CTNNB1/WNT mediated immune evasion in lung cancer. They were able to link Wnt 1 expression with decreased immune activation in human cancer patients and decreased T cell accumulation in lung cancer. Using a mouse model they were able to show that decreased T cell activation was due to altered DC pool mediated via paracrine Wnt1. Blocking this pathway reversed the observed phenotype and thus presents a therapeutic opportunity in lung cancer.

Several concerns remain:

Figure 1 the authors argument is that Wnt 1 is negatively correlated with T cell infiltration, which they show for lung cancer. However it would be advantageous if they could show if this applies to all analyzed cancers or is lung cancer specific.

Figure 2 the analysis of immune infiltrate is only superficial. The authors should report absolute numbers instead of percentage to reflect a true reduction. This could be accompanied by IHC or IF staining to prove failed infiltration.

The biggest concern here is that the authors use a very rough grouping of DC although their main argument is a direct effect on the phenotype. They should at least investigate DC1, DC2, moDC, pDC and langerhans cells. As they globally delete all DC later in Figure 4 they should know that all cDC are equally affected. There is an increasing amount of literature showing that chemokines and receptors are differentially expressed between DC subsets thus knowing the composition of DC might be highly informative.

Figure 5 In addition to chemokines are the corresponding receptors also affected? How about MHC I expression?

General the authors build a convincing argument for Wnt1 acting on DC however there is a shortcoming in linking altered chemokine expression with reduced T cell priming? Does this occur in the LN or in the tumor? Can they use blocking antibodies in their in vitro assay to show that reduced chemokines result in reduced priming?

Reviewer #3:

Remarks to the Author:

In this manuscript titled "Wnt1 silences CC/CXC motif chemokine genes in dendritic cells and induces adaptive immune resistance in lung adenocarcinoma" by Kerdidani et al., have investigated the paracrine effect Wnt1 on antitumor immune responses. Using TCGA transcriptomics database, authors have identified Wnt1 is highly expressed in human LUADs and this was inversely associated T cells abundance. As consequence, altering Wnt1 expression by LLC tumor cells markedly affected the growth tumor growth and this was dependent on DCs and T cells. By performing RNAseq on intratumoral DCs, authors have identified that paracrine Wnt1 signaling in DCs results in silencing of CC/CXC chemokine expression and this is associated with impaired cross-priming of CD8+ T cells. Further, authors have demonstrated that blocking Wnt1 expression alone or with vaccination markedly enhanced antitumor immune responses with reduced tumor growth and burden in the lung. Several studies have shown the paracrine effect of Wnts on antitumor immune responses in several models of transplantable tumors. Key finding observed with present study is the identification of paracrine Wnt1 signaling in DCs silences of CC/CXC chemokine in the tumors. Even though the study is interesting and many observations are novel, there are several major concerns with this study that need to be addressed. In addition, some technical questions and a disjointed set of analysis temper enthusiasm for the current draft.

1. Most the experiments and interpretations are based on using one tumor cell line (LLC). It would be interesting to see several of the interesting observations made in the present study is applicable other tumors. This would further strengthen the current findings by using another lung cancer tumor cell line and spontaneous lung cancer mouse model.
2. It is well established that tumor microenvironment (TME) contains high levels of several Wnt ligand. In addition to tumor cells, tumor infiltrating macrophages also express high levels of Wnts. Authors need show expression of levels of various Wnt ligands by LLC tumors.
3. It would be interesting to see whether antitumor immune responses observed in the present study is specific to Wnt1 or overexpression of any Wnt ligand that activate b-catenin pathway would have similar effect on antitumor responses in the lung.
4. Data showing link between paracrine Wnt1-signaling in intratumoral DCs and activation of b-catenin is weak. In addition to Wnts, multiple signaling pathways activate b-catenin. Authors should perform ex vivo study to test whether ex vivo treatment of splenic DCs with rWnt1 activates b-catenin (Western blot, FACS).
5. Representing data in frequency (%) of immune cells infiltrating tumors might be misleading, as this might change depending on the tumor burden and size. Authors should represent these data as total number of specific immune cells (eg Fig 2E, G). This should be applied to other figures through the MS
6. Figure 4A data should also be represented as MFI and number of DCs positive for active b-catenin. Representative FACS plot with isotype control should be shown as supplementary data. Authors should also look at the b-catenin activation status in DCs in draining lymph node to support that activation specifically happens in the TME and is mediated by Wnt1. Figure 4B data should also be represented as MFI and number of DCs positive for b-gal in TME and DLN.
7. Since DCs and macrophages can uptake the liposomes, authors should the rule out possibility off target effect of Wnt1 shRNA and observed phenotype is the effect of targeting Wnt1 specifically in tumor cells. Fig S1 show that LLC express significant levels of Wnt1 under steady state. Like experiment E, authors should perform similar experiment using LLC tumors without Wnt1 over expression. Authors should also quantify the levels of Wnt1 (ELISA) in mice with LLC tumors and LLC tumors overexpressing Wnt1.
8. Figure legends needs to be more descriptive as it is difficult to follow through the experiments and logical flow in the result sections. (eg. Figure 2B) lung tumor burden ($\times 10^6$) represents tumor nodules or cell. How the tumor burden was calculated should be described in methods sections.

Point-by-Point Reply

REVIEWER 1

Comment 1: *Figure 1 the authors argument is that Wnt 1 is negatively correlated with T cell infiltration, which they show for lung cancer. However it would be advantageous if they could show if this applies to all analyzed cancers or is lung cancer specific.*

Response: We appreciate this valid comment by the reviewer. The exact studies (immunohistochemistry for T cell numbers and qPCR for Wnt1 gene expression) cannot be performed prospectively within a review period. At least 1 year would be needed from getting IRB approval, to acquiring specimens from other types of cancer and analyzing the samples. Therefore, we sought for readily available T cell enumeration and gene expression data. We searched in PubMed, Google and presented abstracts during the last 3 years' oncology/immunooncology scientific congresses, for studies showing transcriptomics and T cell IHC data from non-lung primary tumors. We found one group that had performed such analyses in an adequate number of colorectal tumors that would give us statistical power to detect an existing correlation¹. Considering that Wnt1 is a negative prognostic factor in colorectal cancer, we expected a negative correlation². After communicating personally with Dr Tan Bee Huat Iain, he agreed to provide us a confidential graph depicting intratumoral CD8+ T cell numbers (IHC) versus normalized Wnt1 raw counts (RNAseq), presented below.

Figure 1. In primary colon adenocarcinomas, intratumoral T cytotoxic cells do not correlate with Wnt1 gene expression. Paired tumor samples from colon adenocarcinomas (n=64) were analyzed for T cell infiltration by IHC and Wnt1 gene expression by RNAseq. T cytotoxic cells were expressed as square root normalized number of CD8+ve cells per total number of cells.

These unpublished data clearly show no correlation between Wnt1 gene expression and CD8+ T cell numbers in colorectal cancer (*see p6*).

Having been able to acquire IHC/transcriptomics data for only one non-lung cancer type, we explored alternative approaches to investigate whether Wnt1 negatively correlates to T cells in more cancers. The abundance of CD8⁺ T cells can be measured by expression of the signature genes CD8a and CD8b³. Therefore, we assessed the distribution of Wnt1-CD8a, Wnt1-CD8b correlations in the TCGA database, including paired tumor and tumor-free samples, by calculating the z-scores of the spearman correlation values. We focused on breast, hepatocellular, gastric and clear cell renal carcinomas, because they had been previously associated with Wnt1⁴⁻⁷. We used LUAD as positive control and

colon carcinoma as negative control. GTEx database was also used in order to include an unrelated to TCGA control. LUAD was the only tumor type that the z-score was negative for both CD8a and CD8b (see p6 and Fig S1). Additionally, LUAD was the tumor type with the lowest Wnt1-CD8A correlation z-score. Albeit these results cannot safely rule out a negative correlation between Wnt1 and T cells in other tumors, they suggest a more important role for Wnt1 in LUAD tolerance. This could be related to Wnt1 overexpression being particularly frequent in LUAD compared to other types of cancer (see p6).

Comment 2: *Figure 2 the analysis of immune infiltrate is only superficial. The authors should report absolute numbers instead of percentage to reflect a true reduction. This could be accompanied by IHC or IF staining to prove failed infiltration.*

Response: We thank the reviewer for pointing this out. To confirm that we observe a true and not relative reduction in endogenous and adoptively transferred T cells we have performed new experiments and included FACS beads upon data acquisition to enumerate absolute numbers of T cells. To avoid tumor size-driven bias, numbers of tumor-infiltrating leukocytes can be presented relative to tumor volume, weight or cancer cells^{8,9}. For increased accuracy, we quantified absolute numbers of cancer cells and we depict in our graphs immune cells per cancer cell. Our new analyses convincingly show that Wnt1 decreases endogenous and adoptively transferred T cells (see Fig. 2). Representative IF staining for CD8 further supports the decrease in CD8⁺ cells in Wnt1 overexpressing tumors (see Fig. S5).

Comment 3: *The biggest concern here is that the authors use a very rough grouping of DC although their main argument is a direct effect on the phenotype. They should at least investigate DC1, DC2, moDC, pDC and langerhans cells. As they globally delete all DC later in Figure 4 they should know that all cDC are equally affected. There is an increasing amount of literature showing that chemokines and receptors are differentially expressed between DC subsets thus knowing the composition of DC might be highly informative.*

Response: Recent works have highlighted DC heterogeneity and plasticity in response to stimuli, but also division of labor between DC subsets. We, therefore, thank the reviewer for giving us the opportunity to investigate whether Wnt1 overexpression affects the composition of DC populations. Two independent experiments showed that Langerhans cells do not infiltrate orthotopic LLC tumors. The other 4 subsets, i.e. cDCs1, cDCs2, pDCs and moDCs, were consistently identified, with cDCs1 and pDCs being particularly scarce. No differences were detected between Wnt1 overexpressing and control tumors in either experiment (see p17 and Fig. S7).

Comment 4: *Figure 5 In addition to chemokines are the corresponding receptors also affected? How about MHCI expression?*

Response: We have assessed chemokine receptor expression in DCs and T cells by FACS analysis and detected no differences between Wnt1 overexpressing and control tumors (see p23 and Fig. S13). This was in line with our DC RNAseq data (data not shown). Strikingly, MHCI protein (FACS) was found decreased in cDCs of Wnt1 tumors

versus controls (*see* p23 and Fig. S14). This was paralleled by a nearly significant decrease in MHCI transcripts (FDR=0.0052). Therefore, in addition to chemokines, the Wnt/b-catenin pathway represses MHCI genes in DCs. MHCI downregulation may act synergistically to decreased chemokine secretion against T cell priming by DCs. We thank the reviewer for prompting us to address MHCI expression and strengthen our studies.

Comment 5: *General the authors build a convincing argument for Wnt1 acting on DC however there is a shortcoming in linking altered chemokine expression with reduced T cell priming? Does this occur in the LN or in the tumor?*

Response: Wnt proteins may signal in a short or long-range, upon release by solubilization¹⁰, formation of exosomes^{11,12} or loading on lipid protein particles¹³. It is unknown whether long-range Wnt signaling can occur in LUAD. We assessed Wnt pathway activation in mesothoracic lymph node cDCs of Axin2 reporter mice. We detected no differences between mice bearing Wnt1 overexpressing or control lung tumors (*see* p18 and Fig. S9). We conclude that Wnt1-induced signaling rather occurs in the tumor microenvironment than in lymph nodes. We also assessed T cell activation in lymph nodes and again we found no differences (data not shown).

Comment 6: *Can they use blocking antibodies in their in vitro assay to show that reduced chemokines result in reduced priming?*

Response: We have repeated our DC-T cell co-culture assays in the presence of chemokine blocking antibodies, as requested by the reviewer. Blocking DC-derived chemokines (CCL3, CCL4, CXCL9, CXCL10, CXCL11) hindered T cell priming, confirming dependence of T cell activation on chemokine signaling (*see* p23 and Fig. S15)

REVIEWER 2

Comment 1: *Most the experiments and interpretations are based on using one tumor cell line (LLC). It would be interesting to see several of the interesting observations made in the present study is applicable other tumors. This would further strengthen the current findings by using another lung cancer tumor cell line and spontaneous lung cancer mouse model.*

Response: We agree that our studies were limited by the use of only one lung cancer cell line and transplantation models of lung cancer. To respond to this comment, first we transduced another lung adenocarcinoma cell line, derived from an autochthonous urethane-induced lung tumor, i.e. the Fula cell line¹⁴, with Wnt1 viral vector. Implantation of Wnt1 overexpressing versus control Fula cells in the lungs of syngeneic mice resulted in faster tumor growth and immune evasion (*see* p10 Fig. S6). We also validated Wnt1 as immunotherapeutic target in autochthonous lung adenocarcinomas. Although Kras mutant genetically engineered mouse models are commonly used to test novel therapeutic targets, kras mutant lung adenocarcinoma cells express low Wnt1¹⁵. This was confirmed by our own preliminary experiments (FACS and IHC, data not shown). By contrast, Wnt1 was higher in urethane-induced lung tumors compared to

healthy lungs (Fig. 7A). We therefore treated mice with established urethane-induced lung adenocarcinomas with siWnt1-loaded versus control liposomes. siWnt1 treatment reduced tumor burden, accompanied by increased numbers of T cytotoxic cells and decreased infiltration by b-catenin active cDCs (Fig. 7B).

Comment 2: *It is well established that tumor microenvironment (TME) contains high levels of several Wnt ligand. In addition to tumor cells, tumor infiltrating macrophages also express high levels of Wnts. Authors need show expression of levels of various Wnt ligands by LLC tumors.*

Response: Indeed, the TME contains high levels of several Wnt ligands. Gene expression analysis at the RNA level confirmed that LLC cells express variable levels of several Wnt ligands in vitro and in vivo (see p9 Fig S2).

Comment 3: *It would be interesting to see whether antitumor immune responses observed in the present study is specific to Wnt1 or overexpression of any Wnt ligand that activate b-catenin pathway would have similar effect on antitumor responses in the lung.*

Response: Lung cancers are characterized by dysregulation of Wnt ligand transcription rather than Wnt pathway mutations. Albeit Wnt1 is most frequently overexpressed and serves as a strong negative prognostic factor in LUAD, other canonical Wnts have been also found overexpressed¹⁶⁻²⁰. Among these, Wnt3a is the prototype canonical Wnt ligand. In hepatocellular and colon tumors, T cells and myeloid-like cells produce Wnt3a, which inhibits T cell differentiation towards effector cells^{21,22}. In addition, blocking Wnt3a antibody, administered in vivo, increases expression of the activation marker OX40L in tumor-infiltrating DCs²². We therefore explored whether inducing Wnt3a overexpression in LLC cells might impact antitumor responses in the lung (see p20 Fig. S11). Wnt3-overexpressing tumors showed a significant growth advantage in vivo, compared to Empty tumors. However, T cells were not excluded from tumors. CD44 was relatively low in intratumoral T cells, which may be due to direct Wnt3a-induced suppression²¹. Therefore, the immunological profile of Wnt3a overexpressing LUADs seems to be independent of direct cDC signaling.

Comment 4: *Data showing link between paracrine Wnt1-signaling in intratumoral DCs and activation of b-catenin is weak. In addition to Wnts, multiple signaling pathways activate b-catenin. Authors should perform ex vivo study to test whether ex vivo treatment of splenic DCs with rWnt1 activates b-catenin (Western blot, FACS).*

Response: We agree that ex-vivo exposure of DCs to rWnt1 would strengthen our data on Wnt1-induced signaling on DCs. There are important hurdles in manufacturing active recombinant Wnts. Wnt ligands are modified post-translationally by palmitoylation, which is essential for their function and interaction with FZD receptors. As a result of their acylation, Wnts are very hydrophobic and require detergents for purification, which presents major obstacles to the preparation of active recombinant Wnt proteins²³. We acquired a commercially available rWnt1 protein and recapitulated experimental conditions under which rWnt3a triggers signaling^{21,24}. rWnt1 succeeded to activate b-catenin in purified splenic DCs (see p18 sup Fig. S10). To substantiate further Wnt1 paracrine signaling in DCs, we exposed splenic DCs in culture supernatants of several

Wnt1 over-expressing cancer cells versus those of control cells. We consistently observed b-catenin activation upon exposure to Wnt1 cell-derived versus control supernatants (Fig. S10). Taken together these data they strongly support the link between paracrine Wnt1-signaling and activation of b-catenin in DCs.

Comment 5: *Representing data in frequency (%) of immune cells infiltrating tumors might be misleading, as this might change depending on the tumor burden and size. Authors should represent these data as total number of specific immune cells (eg Fig 2E, G). This should be applied to other figures through the MS.*

Response: We thank the reviewer for pointing this out. To confirm that we observe a true and not relative reduction in endogenous and adoptively transferred T cells we have performed new experiments and included FACS beads upon data acquisition to enumerate absolute numbers of T cells. To avoid tumor size-driven bias, numbers of tumor-infiltrating leukocytes can be presented relative to tumor volume, weight or cancer cells^{8,9}. For increased accuracy, we quantified absolute numbers of cancer cells and we depict in our graphs immune cells per cancer cell. Our new analyses convincingly shows that Wnt1 decreases endogenous and adoptively transferred T cells (see Fig. 2). Representative IF staining for CD8 further supports the decrease in CD8⁺ cells in Wnt1 overexpressing tumors (see Fig. S5).

Comment 6: *Figure 4A data should also be represented as MFI and number of DCs positive for active b-catenin. Representative FACS plot with isotype control should be shown as supplementary data. Authors should also look at the b-catenin activation status in DCs in draining lymph node to support that activation specifically happens in the TME and is mediated by Wnt1. Figure 4B data should also be represented as MFI and number of DCs positive for b-gal in TME and DLN.*

Response: We have added MFIs, numbers of DCs and FACS plots with controls, as requested (Fig. 4). It is unknown whether long-range Wnt signaling can occur in LUAD. We assessed b-catenin activation and b-galactosidase expression in mesothoracic lymph node DCs of Wnt pathway reporter mice bearing Wnt1 overexpressing tumors. Albeit Wnt proteins may signal in a long-range, upon release by solubilization¹⁰, formation of exosomes^{11,12} or loading on lipid protein particles¹³, we detected no differences to mice bearing control tumors (see p18 and Fig. S9). We conclude that Wnt1-induced signaling rather occurs in the TME than in lymph nodes.

Comment 7: *Since DCs and macrophages can uptake the liposomes, authors should rule out possibility of off target effect of Wnt1 shRNA and observed phenotype is the effect of targeting Wnt1 specifically in tumor cells. Fig S1 show that LLC express significant levels of Wnt1 under steady state. Like experiment E, authors should perform similar experiment using LLC tumors without Wnt1 over expression. Authors should also quantify the levels of Wnt1 (ELISA) in mice with LLC tumors and LLC tumors overexpressing Wnt1.*

Response: We have ruled out the possibility of off target effect of Wnt1 siRNA nanoparticles in DCs and macrophages, by assessing Wnt1 expression by FACS. Neither cell type expressed Wnt1 at the protein level (data not shown). In accordance, our

purified intratumoral cDCs did not show Wnt1 gene expression by RNA sequencing (GSE123068) (*see* p26). We also questioned whether siWnt1 nanoparticles might act therapeutically against wild-type LLC tumors. We did observe a less impressive, but still statistically significant response (*see* p27 Fig.S16). Wnt1 protein levels (ELISA and Western) were higher in vivo in Wnt1 overexpressing versus control LLC tumors (*see* p8 Fig.S4).

Comment 8: *Figure legends needs to be more descriptive as it is difficult to follow through the experiments and logical flow in the result sections. (eg. Figure 2B) lung tumor burden ($X10^6$) represents tumor nodules or cell. How the tumor burden was calculated should be described in methods sections.*

Response: We have included additional information in most figure legends. We hope it makes it easier for the reader to follow the experimental flow. We have also added details on tumor burden calculations (*see* p48).

REFERENCES

1. Clarinda Wei Chua, E.C., Elisa Fontana, Si Lin Koo, Joe Poh Yeong, Andy Nguyen, J. Zachary Sanborn, Steve Benz, Emile John Tan, Ronnie Mathew, Ee-Lin Toh, Sarah Boon Ng, Tony Kiat Lim, Anders Jacobsen Skanderup, Shahrooz Rabizadeh, Anguraj Sadanandam, Jonathan Göke and Iain Bee Tan Abstract 5693: Tumor whole-transcriptome sequencing and multiplex immunohistochemistry of immune cell populations in 158 Asian colorectal cancers. *Cancer Research Proceedings: AACR Annual Meeting 2018*; April 14-18, 2018; Chicago, IL.(2018).
2. Stanczak, A., *et al.* Prognostic significance of Wnt-1, beta-catenin and E-cadherin expression in advanced colorectal carcinoma. *Pathol Oncol Res* **17**, 955-963 (2011).
3. Danaher, P., *et al.* Gene expression markers of Tumor Infiltrating Leukocytes. *J Immunother Cancer* **5**, 18 (2017).
4. Zardawi, S.J., O'Toole, S.A., Sutherland, R.L. & Musgrove, E.A. Dysregulation of Hedgehog, Wnt and Notch signalling pathways in breast cancer. *Histol Histopathol* **24**, 385-398 (2009).
5. Zhang, J.G., *et al.* MiR-148b suppresses cell proliferation and invasion in hepatocellular carcinoma by targeting WNT1/beta-catenin pathway. *Sci Rep* **5**, 8087 (2015).
6. Cha, Y., *et al.* MicroRNA-140-5p suppresses cell proliferation and invasion in gastric cancer by targeting WNT1 in the WNT/beta-catenin signaling pathway. *Oncol Lett* **16**, 6369-6376 (2018).
7. Kruck, S., *et al.* Impact of an altered Wnt1/beta-catenin expression on clinicopathology and prognosis in clear cell renal cell carcinoma. *Int J Mol Sci* **14**, 10944-10957 (2013).
8. Salmon, H., *et al.* Expansion and Activation of CD103(+) Dendritic Cell Progenitors at the Tumor Site Enhances Tumor Responses to Therapeutic PD-L1 and BRAF Inhibition. *Immunity* **44**, 924-938 (2016).

9. Green, J.A., Arpaia, N., Schizas, M., Dobrin, A. & Rudensky, A.Y. A nonimmune function of T cells in promoting lung tumor progression. *J Exp Med* **214**, 3565-3575 (2017).
10. Mulligan, K.A., *et al.* Secreted Wingless-interacting molecule (Swim) promotes long-range signaling by maintaining Wingless solubility. *Proc Natl Acad Sci U S A* **109**, 370-377 (2012).
11. Gross, J.C., Chaudhary, V., Bartscherer, K. & Boutros, M. Active Wnt proteins are secreted on exosomes. *Nat Cell Biol* **14**, 1036-1045 (2012).
12. Luga, V., *et al.* Exosomes mediate stromal mobilization of autocrine Wnt-PCP signaling in breast cancer cell migration. *Cell* **151**, 1542-1556 (2012).
13. Neumann, S., *et al.* Mammalian Wnt3a is released on lipoprotein particles. *Traffic* **10**, 334-343 (2009).
14. Agalioti, T., *et al.* Mutant KRAS promotes malignant pleural effusion formation. *Nat Commun* **8**, 15205 (2017).
15. Tammela, T., *et al.* A Wnt-producing niche drives proliferative potential and progression in lung adenocarcinoma. *Nature* **545**, 355-359 (2017).
16. Huang, C.L., *et al.* Wnt1 overexpression promotes tumour progression in non-small cell lung cancer. *European journal of cancer* **44**, 2680-2688 (2008).
17. Stewart, D.J. Wnt Signaling Pathway in Non-Small Cell Lung Cancer. *Jnci-Journal of the National Cancer Institute* **106**(2014).
18. Lindskog, C., Edlund, K., Mattsson, J.S. & Micke, P. Immunohistochemistry-based prognostic biomarkers in NSCLC: novel findings on the road to clinical use? *Expert review of molecular diagnostics* **15**, 471-490 (2015).
19. Xu, X., *et al.* Immunohistochemical demonstration of alteration of beta-catenin during tumor metastasis by different mechanisms according to histology in lung cancer. *Experimental and therapeutic medicine* **9**, 311-318 (2015).
20. Xu, X., *et al.* Aberrant Wnt1/beta-catenin expression is an independent poor prognostic marker of non-small cell lung cancer after surgery. *Journal of thoracic oncology : official publication of the International Association for the Study of Lung Cancer* **6**, 716-724 (2011).
21. Schinzari, V., *et al.* Wnt3a/beta-Catenin Signaling Conditions Differentiation of Partially Exhausted T-effector Cells in Human Cancers. *Cancer Immunol Res* **6**, 941-952 (2018).
22. Pacella, I., *et al.* Wnt3a Neutralization Enhances T-cell Responses through Indirect Mechanisms and Restrains Tumor Growth. *Cancer Immunol Res* **6**, 953-964 (2018).
23. Janda, C.Y., *et al.* Surrogate Wnt agonists that phenocopy canonical Wnt and beta-catenin signalling. *Nature* **545**, 234-237 (2017).
24. Oderup, C., LaJevic, M. & Butcher, E.C. Canonical and noncanonical Wnt proteins program dendritic cell responses for tolerance. *J Immunol* **190**, 6126-6134 (2013).

Reviewers' Comments:

Reviewer #2:

Remarks to the Author:

The authors have successfully addressed some of the concerns of the reviewer. However it remains to be address which DC and DC populations are affected. It is somewhat unclear to the reviewer why in figure S 17 the authros show human Dc phenotyping while working in a murine model. They should phenotype and report the DC subsets observed in their mouse model. Further the experiments in the revision suggest an Wnt1 effect on Dc MHC-I expression. This can be seen as a surrogate for Dc activation. What is the activation status of the DC? Are they equally capable of cross-presentation? These possibilities need at least to be discussed at length if not addressed.

Lastly the authors fail to show where the T cell priming is occurring in vivo? Is trafficking tot he LN needed or is the priming occurring within the tumor itself.

Reviewer #3:

Remarks to the Author:

In the revised submission, the authors have performed additional experiments addressing the previous concerns raised. With the additional data, the manuscript is convincing and further, strengthen the conclusions.

Point-by-Point Reply

REVIEWER 2

Comment 1: *However it remains to be address which DC and DC populations are affected. It is somewhat unclear to the reviewer why in figure S 17 the authros show human Dc phenotyping while working in a murine model. They should phenotype and report the DC subsets observed in their mouse model.*

Response: We appreciate this valid comment by the reviewer. We have reported the DC subsets observed in our mouse model and there were no differences between Wnt1 overexpressing and control tumors (see p17 and Fig. S7). Therefore, the observed cDC defects are not due to alterations in the composition of the cDCs. Moreover, active b-catenin did not differ between intratumoral cDCs1 and cDCs2 of Wnt1-LLC tumors (data not shown), suggesting similar levels of Wnt pathway activation on both subsets. We have added this information to the Results section (see p17). It would also be interesting to phenotype cDC subsets, e.g. investigate whether Wnt1 down-regulates chemokines in each subset, but we feel that it falls outside the scope of this study. In addition, in our murine model cDCs1 are particularly rare, as reported by us and others (Oncoimmunology. 2017; 6(1): e1253655). This hampers their purification for downstream analysis. We were able, however, to purify cDCs1 from human tumors, as they were less scarce. Additionally, all the functional experiments (DC adoptive transfers, purification and ex vivo immunological assays, DC knock-out model) that are shown were based on total cDCs. We have acknowledged the significance of analyzing cDC subsets and suggested it as a topic for further research in the Discussion section (see p33).

Comment 2: *“Further the experiments in the revision suggest an Wnt1 effect on Dc MHC-I expression. This can be seen as a surrogate for Dc activation. What is the activation status of the DC? Are they equally capable of cross-presentation? These possibilities need at least to be discussed at length if not addressed.”*

Response: The activation status of intratumoral cDCs of Wnt1 overexpressing versus control lung tumors was assessed via gene expression and functional analysis. RNAseq did not show a difference in expression of co-stimulatory or co-inhibitory molecules. Additionally, neither IL-12 nor TNF α were found decreased in culture supernatants of stimulated purified cDCs from Wnt1 tumors. The RNAseq data also showed no defects in expression of genes that regulate cross-presentation pathways. Albeit downregulation of MHCI was expected to negatively impact the cross-presenting ability of the cDCs, Wnt1-exposed cDCs were capable of cross-presenting mCherry in vivo. However, mCherry cross-presentation may not necessarily recapitulate cross-presentation of other cancer antigens. We have discussed these possibilities in the revised manuscript (see p33-34).

Comment 3: *Lastly the authors fail to show where the T cell priming is occurring in vivo? Is trafficking tot he LN needed or is the priming occurring within the tumor itself.*

Response: Wnt1-induced signaling on cDCs occurs in the tumor microenvironment rather than regional lymph nodes, as there was no evidence of Wnt pathway activation in nodal cDCs (*see*

p18 and Fig. S9) and we detected no difference in numbers and CD44 expression by T cells infiltrating the mesothoracic lymph nodes of Wnt1 overexpressing versus control tumors (data not shown). We have clarified our explanation in the Discussion section to highlight that reduced chemokine expression by intratumoral cDCs likely impacts effector T cells trafficking and priming at tumor sites (see p32).

Reviewers' Comments:

Reviewer #2:

Remarks to the Author:

All concerns have been addressed.

Point-by-Point Reply

REVIEWER 2

Comment 1: All concerns have been addressed.

Response: We thank the reviewer for this fruitful review process.